# Natural ITD statistics predict human auditory spatial perception

**Rodrigo Pavão[1,2]\*, Elyse S Sussman[1], Brian J Fischer[3], José L Peña[1]**

[1]Dominick P. Purpura Department of Neuroscience - Albert Einstein College of Medicine, New York, United States; [2]Centro de Matemática, Computação e Cognição - Universidade Federal do ABC, Santo André, Brazil; [3]Department of Mathematics - Seattle University, Seattle, United States

**Abstract** A neural code adapted to the statistical structure of sensory cues may optimize perception. We investigated whether interaural time difference (ITD) statistics inherent in natural acoustic scenes are parameters determining spatial discriminability. The natural ITD rate of change across azimuth (ITDrc) and ITD variability over time (ITDv) were combined in a Fisher information statistic to assess the amount of azimuthal information conveyed by this sensory cue. We hypothesized that natural ITD statistics underlie the neural code for ITD and thus influence spatial perception. To test this hypothesis, sounds with invariant statistics were presented to measure human spatial discriminability and spatial novelty detection. Human auditory spatial perception showed correlation with natural ITD statistics, supporting our hypothesis. Further analysis showed that these results are consistent with classic models of ITD coding and can explain the ITD tuning distribution observed in the mammalian brainstem.

**\*For correspondence:** rodrigo.pavao@ufabc.edu.br

**Competing interests:** The authors declare that no competing interests exist.

## Introduction

Humans and other species localize sound sources in the horizontal plane using sub-millisecond interaural time difference (ITD) between signals arriving at the two ears (*Middlebrooks and Green, 1991*). ITD is detected by auditory brainstem neurons within narrow frequency bands (*Goldberg and Brown, 1969*; *Yin and Chan, 1990*; *Carr and Konishi, 1990*; *McAlpine et al., 2001*).

Classical psychophysical studies demonstrated that humans detect sound location better in the front than in the periphery (*Mills, 1958*; *Yost, 1974*; *Makous and Middlebrooks, 1990*). Enhanced performance at frontal locations could be efficient for hunting and foraging, as proposed for vision (*Collins and Opthalmological Society of the United Kingdom, 1922*; *Cartmill, 1974*; *Changizi and Shimojo, 2008*). Physiological evidence indicates that coding of binaural spatial cues could support finer discriminability in the front (*van Bergeijk, 1962*; *Feddersen et al., 1957*; *McAlpine et al., 2001*; *Grothe et al., 2010*). Better sound discrimination and localization in frontal locations can also be predicted from the geometry of the head and the placing of the ears, causing higher ITD rate of change as a function of azimuth in the front (*Woodworth, 1938*; *Feddersen et al., 1957*; *Gelfand, 2016*).

For azimuth detection based on ITD, sound diffraction by structures surrounding the ear can affect the ITD-azimuth relationship (*Aaronson and Hartmann, 2014*; *Roth et al., 1980*). In addition, because the brain computes ITD in narrow frequency bands, the interaction of nearby frequencies within a given cochlear filter may also be a source of ITD variability over time (ITDv). Sensory variability is known to be fundamental in change detection. It has been shown that stimulus discrimination depends not only on the mean difference between stimuli but also on the variability of the sensory evidence (*Green and Swets, 1966*).

**eLife digest** When a person hears a sound, how do they work out where it is coming from? A sound coming from your right will reach your right ear a few fractions of a millisecond earlier than your left. The brain uses this difference, known as the interaural time difference or ITD, to locate the sound.

But humans are also much better at localizing sounds that come from sources in front of them than from sources by their sides. This may be due in part to differences in the number of neurons available to detect sounds from these different locations. It may also reflect differences in the rates at which those neurons fire in response to sounds. But these factors alone cannot explain why humans are so much better at localizing sounds in front of them.

Pavão et al. showed that the brain has evolved the ability to detect natural patterns that exist in sounds as a result of their location, and to use those patterns to optimize the spatial perception of sounds. Pavão et al. showed that the way in which the head and inner ear filter incoming sounds has two consequences for how we perceive them. Firstly, the change in ITD for sounds coming from different sources in front of a person is greater than for sounds coming from their sides. And secondly, the ITD for sounds that originate in front of a person varies more over time than the ITD for sounds coming from the periphery. By playing sounds to healthy volunteers while removing these differences, Pavão et al. found that natural ITD statistics were correlated with a person's ability to tell where a sound was coming from.

By revealing the features the brain uses to determine the location of sounds, the work of Pavão et al. could ultimately lead to the development of more effective hearing aids. The results also provide clues to how other senses, including vision, may have evolved to respond optimally to the environment.

This study tested the hypothesis that the natural ITD statistics are encoded by the brain, determining ITD perception. Using human HRTF databases and models of cochlear filters, we estimated ITD rate of change and ITDv, and tested whether these statistics combined in a Fisher information metric predicted spatial discrimination thresholds and deviance detection better than ITD rate of change alone. We presented sounds through insert earphones, removing ITD statistics, to determine whether sound location discriminability and spatial deviance detection were predicted by natural ITD statistics independently from the actual stimulus properties. We found that natural ITD statistics were correlated with auditory spatial discriminability and spatial deviance detection. Analysis of classic models of ITD coding (*Stern and Colburn, 1978*; *Harper and McAlpine, 2004*) support the idea that ITD statistics influence the density distribution of ITD tuning, which may be genetically encoded and conserved across human subjects. Thus, our results are consistent with the hypothesis that human brain evolution has incorporated natural statistics of spatial cues to the neural code underlying auditory spatial perception.

## Results

ITD statistics, specifically, the derivative (rate of change) of the mean ITD over azimuth (ITDrc) and the standard deviation of ITD over time (ITDv) were estimated from human HRTFs and models of cochlear filters. We first tested whether these ITD statistics predict human spatial discrimination thresholds measured under free-field sound stimulation from previously published datasets (*Mills, 1958*) and from data collected using tests specifically designed for measuring ITD discrimination through sounds delivered by earphones. Next, we used EEG and mismatch negativity signals (MMN) to address the question of whether these natural ITD statistics influence ITD deviance detection. Finally, we evaluated the compatibility of ITD statistics with the classic neural models for coding ITD.

### ITD statistics estimated from human HRTFs and properties of cochlear filters

To test the hypothesis that natural ITD statistics influence the neural code underlying sound localization, we estimated ITDrc and ITDv in sounds reaching the ears of human subjects. The method for

estimating the ITD mean and standard deviation (*Figure 1A*), which was applied across locations and frequencies, included: (1) Impulse responses obtained from publicly available human HRTF databases (Listen HRTF database; 51 subjects) were convolved with acoustic signals, which results in modulation of ongoing phase and gain that depends on both frequency and sound direction; (2) Sound signals were filtered using models of human cochlear filters (*Glasberg and Moore, 1990*) (3) Instantaneous phase and interaural phase difference (IPD) was extracted from the resulting signals; (4) The mean and standard deviation of instantaneous IPD was computed and converted to ITD to estimate ITD mean and ITD standard deviation over time, at each azimuth and frequency, across subjects (*Figure 1B*); and (5) ITDrc was calculated as the derivative of mean ITD over azimuth and ITDv was calculated as the standard deviation of ITD over time.

The ITDrc and ITDv statistics were combined to compute the Fisher information in ITD at each location and frequency. Estimation theory has shown that the square root of Fisher information relates to discrimination thresholds (*Abbott and Dayan, 1999*; *Brown et al., 2018*). Thus, the square root of ITD Fisher information ($\sqrt{FI_{ITD}}$) was the ITD statistic used in this study (see Methods section for details), which closely approximates the ITDrc/ITDv ratio, computed at each location and frequency (*Figure 1C-left*). $\sqrt{FI_{ITD}}$ displayed low variability across individuals (*Figure 1C-right*), indicating it constitutes a statistic that is largely invariant across human subjects.

ITDrc is determined by the shape and filtering properties of the head, including diffractions (*Woodworth, 1938*; *Aaronson and Hartmann, 2014*; *Roth et al., 1980*). These features should affect ITDv as well; however, ITDv also depends on phase and gain modulations induced by interaction between neighboring frequencies, within the critical band of single cochlear filters (*Figure 1—figure supplement 1A*). Consistently, the correlation between ITDrc and ITDv was not strong ($r_{Spearman} = -0.41$). These ITD statistics were consistent across broadband signals with different frequency spectra (*Figure 1—figure supplement 1B*). Additionally, we tested the consistency of ITD statistics across environments, comparing statistics estimated from HRTFs recorded in anechoic and reverberant rooms (database available in http://medi.uni-oldenburg.de/hrir; *Kayser et al., 2009*). Echoes significantly disrupted ITD statistics; however, the precedence effect is expected to segregate leading signals from their lagging echoes (*Wallach et al., 1949*; *Brown et al., 2015*). Accordingly, ITD statistics estimated in anechoic and reverberant environments were similar when signal transients were considered (*Figure 1—figure supplement 1C*). Finally, the estimated ITDv was equivalent to the variability over trials (obtained from one instantaneous sample of 200 different broadband signals; $r_{Spearman} = 0.99$), indicating that ITDv exhibits ergodicity. Invariance across contexts is a premise motivating the question of whether ITD statistics are represented in the brain, influencing auditory spatial perception.

Different auditory cue statistics have been examined in previous studies, as well as their predictive accuracy of human auditory spatial discriminability. Higher ITDrc in the midline supports the better spatial discrimination observed in frontal locations (*Mills, 1972*; *Gelfand, 2016*; *Brown et al., 2018*). Consistent with previous reports (*Woodworth, 1938*; *Feddersen et al., 1957*; *Gelfand, 2016*), we found that ITDrc was higher in the midline for most frequencies. Additionally, the highest ITDrc occurred at locations distant from the midline in some frequencies, which was also observable in previous studies (*Kuhn, 1977*; *Benichoux et al., 2016*).

Across-trial ITD variability induced by concurrent sounds was reported by *Cazettes et al., 2014* as a metric relevant to the owl's auditory spatial perception. When this method was applied to human HRTFs and human cochlear filters, for the range of frequencies of interest in human ITD detection, we found ITD variability values weakly correlated to those obtained with the method used in the present study. However, a stronger correlation between both metrics was observed for the range of frequencies most relevant to owls' sound localization (above 2000 Hz), suggesting that the effect of concurrent sounds on ITDv may not represent a significant source of ITDv in humans. *Młynarski and Jost, 2014* also estimated auditory cue statistics across environments. However, they did so without reporting the location of sound sources, which restricted the use of their estimated statistics for testing the prediction of sound discrimination across locations.

In the current study, we hypothesized that the neural representation of ITD is influenced by natural ITD statistics so that ITD perception is predicted by natural $\sqrt{FI_{ITD}}$ (*Figure 1D*). To test this hypothesis, we investigated the $\sqrt{FI_{ITD}}$ prediction accuracy of ITD discrimination performance and

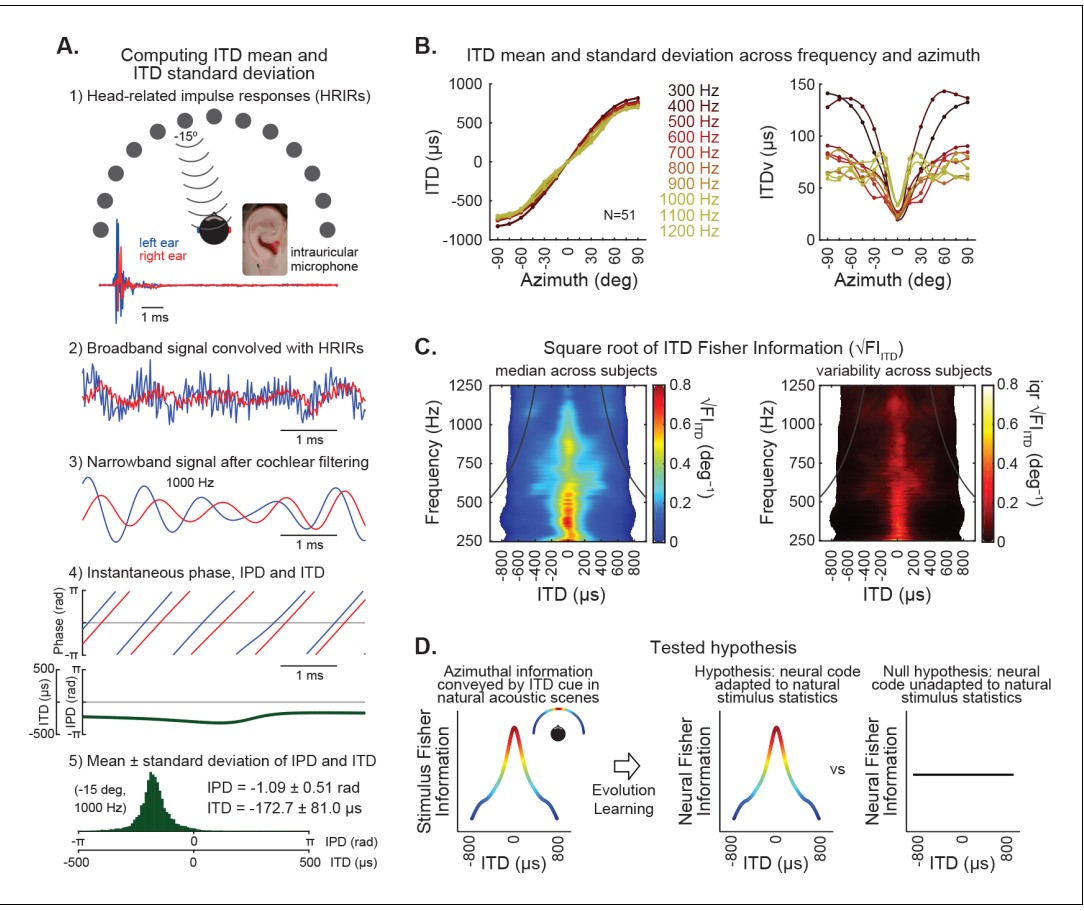

**Figure 1.** ITD statistics of natural stimulus. (**A**) Estimation of ITD mean and standard deviation over time in broadband signals filtered by human head-related impulse responses (HRIRs) and modeled cochlear filters. (1) Example HRIRs from sound emitted from speakers located at −15 degrees and recorded with microphones positioned in each ear (obtained from a publicly available LISTEN dataset). Traces show example impulse responses in the right (red) and left (blue). (2) A broadband signal was convolved with HRIRs from right (red) and left (blue) ears for each direction. (3) Convolved signals were then filtered using parameters analogous to human cochlear filters. Example of signal passed through a cochlear filter with a frequency band centered on 1000 Hz for the left (blue) and right (red) ears. (4) The instantaneous phase of the resulting signals on the left and right ears was computed. Top, instantaneous phase over time for the left (blue) and right (red) signals shown in 3. Bottom, instantaneous phase differences (IPD, in radians) and instantaneous time differences (ITD, in microseconds) between left and right signals. (5) Histogram of instantaneous IPD and ITD, illustrating their variability over time for the example signal shown in 3. (**B**) ITD mean (left) and standard deviation (right) over time, as a function of frequency and azimuth. Plots represent median values across subjects (N = 51), fit by spline curves, and color coded for each frequency. The derivative of the curves on the left was used to calculate ITD rate of change (ITDrc) across azimuth. The ITD variability (ITDv) was computed as the standard deviation of the ITD distribution over time. (**C**) Left, information of ITD cues as a function of frequency and azimuth, quantified by the median square root of ITD Fisher information ($\sqrt{\text{FI}_{\text{ITD}}}$) across subjects (azimuth was converted to ITD to obtain the estimate of the ITD statistics as a function of frequency and ITD, matching the stimulus metrics and model parameters used in our study). $\sqrt{\text{FI}_{\text{ITD}}}$ statistic closely approximates ITDrc/ITDv. Right, the interquartile range of $\sqrt{\text{FI}_{\text{ITD}}}$ across subjects shows low inter-individual variability. Black lines on each panel indicate the $\pi$-limit across frequency, beyond which ITD cues become ambiguous for narrowband sounds. (**D**) This study tests the hypothesis that over evolutionary and/or ontogenetic time scales the human brain became adapted to natural ITD statistics, such that stimuli that are more informative about sound source location would be distinctively encoded.

The online version of this article includes the following figure supplement(s) for figure 1:

**Figure supplement 1.** Basis of ITD statistics and consistency across conditions.

novelty detection. Finally, we evaluated the consistency between ITD statistics and frameworks proposed in two classical neural models for ITD coding.

## Prediction of spatial discrimination thresholds from ITD statistics

A central hypothesis tested by this study was that a neural code adapted to natural ITD statistics influences ITD-change discriminability (dITD) thresholds even under conditions where ongoing stimulus statistics are constant across frequency and locations (*Figure 2A*).

Free-field dITD thresholds as a function of ITD and frequency as reported by a classic study of human sound localization (*Mills, 1958*) were used to test the hypothesis. In addition, a test measuring dITD thresholds through dichotic (earphone) sound delivery was conducted. Neither of these datasets delivered stimuli carrying natural ITD statistics: they both used tones, which abolishes ITDv, and disables ITDrc estimation by either fixing the head of subjects (in free-field) or by decoupling head movement and ITD input (in dichotic). Thus, the effect of natural ITD statistics influencing the neural representation of ITD could be assessed in both approaches.

We first tested whether natural ITD statistics predicted the free-field dITD thresholds estimated from the previously reported dataset (*Mills, 1958*). *Figure 2B* shows the free-field dITD thresholds reported by *Mills, 1958* as a function of ITDrc and $\sqrt{FI_{ITD}}$ estimated in our study. $\sqrt{FI_{ITD}}$ displayed higher correlation with dITD thresholds than ITDrc. These results are consistent with the hypothesis that selectivity for $\sqrt{FI_{ITD}}$ statistic (which combines ITDv and ITDrc) may underlie the evolution of the neural code supporting discrimination thresholds.

Additionally in the current study, to test ITD-change discrimination thresholds and evaluate prediction accuracy of ITD statistics we presented sounds through earphones instead of free-field stimulation. This avoids potential effects of ongoing ITD statistics and the influence of other sound localization cues. The 24 normal-hearing adults that participated in the testing were instructed to detect a change in ITD within a pair of tonal sounds (Methods), which allowed us to obtain dichotic dITD thresholds across reference ITD and frequency, for the range of interest (*Figure 2C-left*). The dichotic dITD thresholds averaged across subjects correlated with free-field thresholds reported by *Mills, 1958* ($r_{Spearman} = 0.64$). However, dITD thresholds were higher in the present study compared to *Mills, 1958*; other studies also using dichotic stimuli reported higher dITD thresholds (e.g. *Brughera et al., 2013*) than *Mills, 1958*. This may be due to approach differences, such as presentation of sounds through earphones vs. free-field stimulation and testing untrained subjects in the present study rather than highly trained individuals as was done in *Mills, 1958*. Using free-field stimulation leaves open the possibility that listeners rely on cues other than ITD to detect sound location in azimuth, which may have lowered the thresholds as found in *Mills, 1958*. Additionally, training in ITD detection may have an effect on threshold levels compared to normal untrained individuals.

We computed the average dichotic dITD thresholds across participants and quantified the Spearman correlation between them and ITD statistics estimated in our study. When all frequency conditions were analyzed, average dichotic dITD thresholds showed moderate correlation with ITDrc (*Figure 2C-middle*) and $\sqrt{FI_{ITD}}$ (*Figure 2C-right*). This was particularly influenced by low correlation for dITD thresholds for 250 Hz tones. Higher thresholds for this frequency have previously been reported (*Brughera et al., 2013*). Exclusion of this frequency substantially improved $\sqrt{FI_{ITD}}$'s prediction accuracy (*Figure 2C-right*), suggesting that dITD thresholds at 250 Hz may be determined by additional parameters not addressed by the ITD statistics investigated in our study. Comparing the prediction accuracy of these statistics using linear mixed-effects models (Materials and methods) resulted in the same outcome as the Spearman correlation analysis. This provides further support for the hypothesis that both ITDrc and ITDv statistics, combined in $\sqrt{FI_{ITD}}$, influence the ITD neural code underlying discrimination thresholds.

## Neural code underlying deviance detection is adapted to ITD statistics

A further investigation to support the idea that ITD statistics are correlated with auditory spatial perception was conducted by testing the ability to detect spatial deviants from a standard sound location in space. To test this, we measured the mismatch negativity (MMN) component of event-related brain potentials (ERPs) (*Näätänen et al., 1978*) for sounds coming from standard (repeated) and deviant (sporadic) spatial locations. MMNs are observable when a sequence of standard stimuli is unexpectedly interrupted by a low probability stimulus, without the listener making an overt

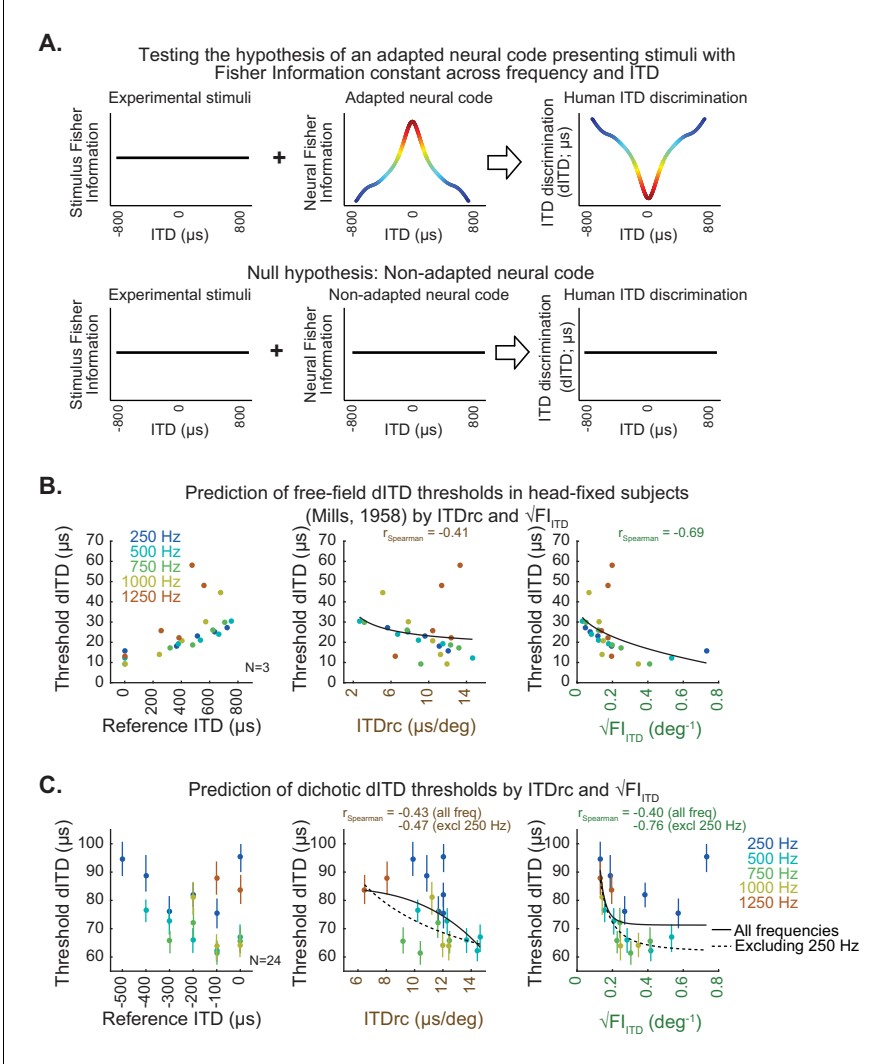

**Figure 2.** ITD statistics predict human ITD-change detection thresholds. (**A**) Hypothesis (top) and null hypothesis (bottom) of an adapted neural code underlying human ITD discrimination. (**B**) Classic study by *Mills, 1958* estimated the minimum azimuth change detection across frequency and locations for sounds in free-field averaged across subjects; these measures were converted to threshold dITD as a function of reference ITD (left). Scatter plots on the middle and right show free-field dITD thresholds as a function of ITDrc and $\sqrt{FI_{ITD}}$. (**C**) Test conducted in the present study to specifically assess dITD thresholds for tonal sounds delivered through headphones (dichotic stimulation). Left, mean dichotic dITD thresholds over subjects as a function of reference ITD across frequency. Middle, dichotic dITD thresholds as a function of ITDrc. Right, dichotic dITD thresholds as a function of $\sqrt{FI_{ITD}}$. Bars indicate 50% confidence intervals of mean dITD thresholds. Black lines represent power functions fit to all the analyzed frequencies (solid) and excluding 250 Hz frequency from the analysis (dotted).

response (*Näätänen et al., 1978*; *Pakarinen et al., 2007*; *Sussman, 2007*; *Sussman et al., 2014*; *Figure 3A-left*). Thus, the MMN signals provide a direct brain measure of discriminability that does not require training subjects to perform behavioral tasks. The MMN signal is displayed by subtracting the mean ERP response elicited by the standard stimuli from the mean ERP elicited by deviant stimuli. The amplitude of the MMN indexes discriminability between standard and deviant sounds. The larger the tone features separation between standard and deviant stimuli, or the larger the perceived difference between standard and deviant, the more negative the MMN amplitude (*Deouell et al., 2006*; *Sams et al., 1985*; *Pakarinen et al., 2007*; *Tiitinen et al., 1994*). Thus, MMN was used to test whether natural ITD statistics influence the magnitude of ITD deviance detection (*Figure 3B*).

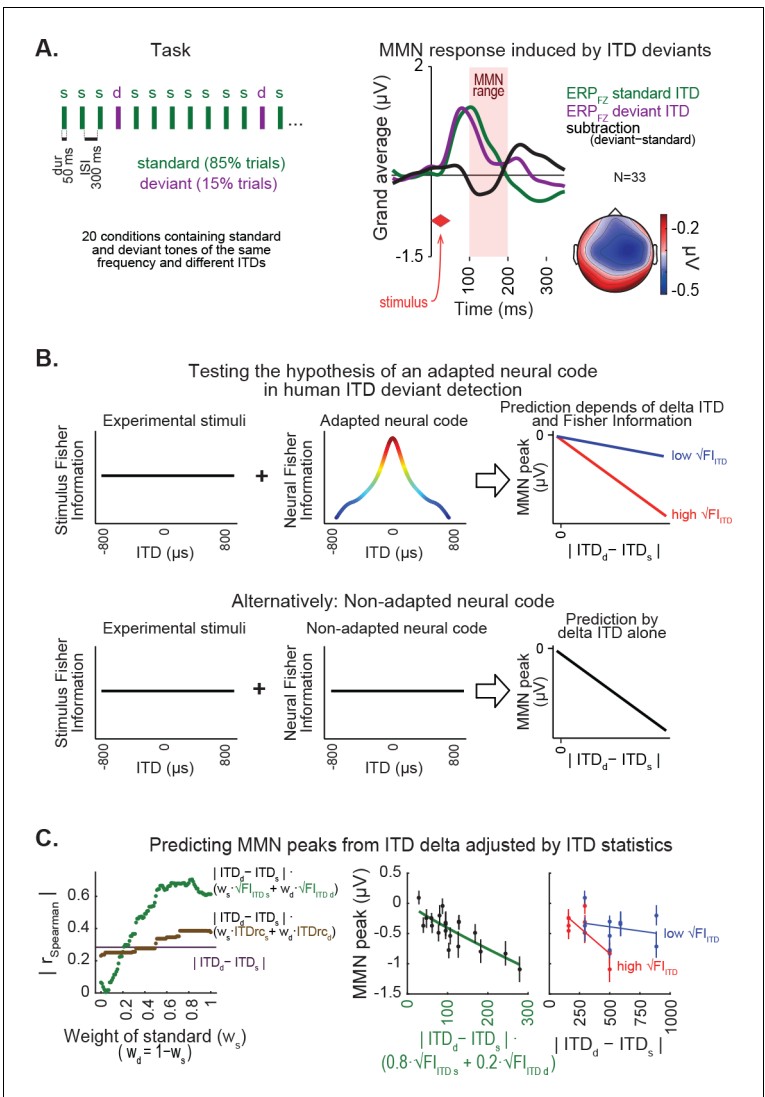

**Figure 3.** ITD statistics predict discriminability of spatial deviants indexed by MMN responses. (**A**) Left, passive oddball sequence protocol, in which subjects listened to frequent 'standard' stimuli embedded with rare spatial 'deviants'. In each condition, two tones were presented with the same frequency and distinct ITDs. Right, MMN response within the 100–200 ms latency range of the deviant-minus-standard trace (black line) is shown for the midline frontal electrode (FZ) along with standard (green) and deviant (purple) event related potential traces, averaged across conditions and subjects. Inset on the bottom-right shows the topography of the MMN response. (**B**) Hypothesis (top) and null hypothesis (bottom) of an adapted neural code underlying MMN responses to spatial deviants tested in this study. Under a neural code relying on natural ITD statistics, the correlation between amplitude of MMN responses and difference between deviant and standard ITD is expected to show a synergistic effect of ITD statistics. (**C**) Left, coefficients of correlations between MMN amplitude and different predictor equations adjusting ITD difference between standard and deviant by ITD statistics, as a function of the relative weight of the standard stimulus ($w_s$), relative to the weight of the deviant ($w_d$). Middle, best prediction of MMN amplitude in the model relying on $\sqrt{FI}_{ITD}$, weighting standard more than deviant (80%:20%). Right panel, changes in MMN peak amplitude as a function of the difference between ITD of deviant and standard show stronger negative linear slopes for conditions where the weighted average of $\sqrt{FI}_{ITD}$ was higher, compared to conditions with lower $\sqrt{FI}_{ITD}$.

A set of ITD and frequency conditions was selected and presented to 33 normal-hearing adults in order to sample critical ranges drawn from the HRTF analysis (Materials and methods). Frequencies of 400, 550, 600 and 650 Hz were chosen because ITDrc and ITDv changed as a function of azimuth in a manner that could maximize the difference in prediction accuracy across ITD statistics. Frequencies lower than 400 Hz were not tested because of observed distortion in the sound stimulation system, while frequencies above 650 Hz were excluded to avoid phase ambiguity confounds. MMN signals were measured separately across participants and conditions. The averaged peak amplitude of MMN was used to quantify the subject's capacity to discriminate between ITDs of the standard and deviant. The characteristic fronto-central scalp topography of the MMN responses were observed (*Giard et al., 1990*; *Figure 3A-right*).

We then examined the prediction accuracy of MMN amplitude of model equations relying on the absolute difference between the ITDs of standard and deviant stimuli adjusted by the weighted sum of ITD statistics of standard and deviant stimuli. The equation we used to test the prediction of MMN amplitude by $\sqrt{FI}_{ITD}$ was:

$$MMNpeak \sim |ITD_d - ITD_s|\left(w_s\sqrt{FI}_{ITD\,s} + (1 - w_s)\sqrt{FI}_{ITD\,d}\right),$$

where $ITD_s$ and $ITD_d$ are the ITD of standard and deviant, $w_s$ and $1-w_s$ are the relative weights of the standard and deviant, and $\sqrt{FI}_{ITDs}$ and $\sqrt{FI}_{ITDd}$ are the estimated $\sqrt{FI}_{ITD}$ values corresponding to the frequency and ITD of the standard and deviant stimuli.

*Figure 3C—left* shows the Spearman correlation between each of the predictors' output and the amplitude of MMN peaks (averaged across subjects) as a function of the weight of the standard. The highest correlation was found when multiplying the ITD difference between standard and deviant by $\sqrt{FI}_{ITD}$, and assigning 80% weight to the standard and 20% to the deviant (*Figure 3C-middle*). Prediction accuracy of model equations using linear mixed-effect models (Materials and methods) yielded the same results as the Spearman correlation analysis. *Figure 3C-right* shows that conditions with higher weighted $\sqrt{FI}_{ITD}$ display larger changes in MMN amplitude as a function of difference between ITD of deviant and standard than conditions with low weighted $\sqrt{FI}_{ITD}$. The good prediction of MMN by the model relying on $\sqrt{FI}_{ITD}$ further supports the idea that combined ITDrc and ITDv are critical in auditory spatial perception.

## Classic neural models of ITD discriminability are consistent with a representation of ITD statistics

Two classic models of neural coding underlying discriminability of azimuth positions in acoustic space based on ITD (*Stern and Colburn, 1978*; *Harper and McAlpine, 2004*) were used to examine the potential link between the brain representation of sensory statistics and perceptual functions. The model by *Stern and Colburn, 1978* postulated an increased density of pairs of fibers underlying tuning to ITDs near the midline, under a labeled-line code framework, as the basis for increased ITD discriminability in the front (*Figure 4A*). This density distribution showed high correlation with $\sqrt{FI}_{ITD}$ (*Figure 4A*) and prediction accuracy of the experimental data of dITD thresholds and ITD deviant detection. Additionally, the density distribution of the model was adjusted to match ITD statistics (*Figure 4B*) by defining the density of cells tuned to each ITD as a linear transformation of $\sqrt{FI}_{ITD}$. The *Stern and Colburn, 1978* model required only minor changes to the density distribution originally proposed to represent the $\sqrt{FI}_{ITD}$ pattern. This indicates that this seminal model, which explains multiple experimental findings, is consistent with a density distribution of ITD tuning influenced by the natural ITD statistics.

On the other hand, the model postulated by *Harper and McAlpine, 2004* relies on the maximization of Fisher information of firing rate within the physiological ITD range to explain the optimal IPD-tuning distribution in a population of brainstem neurons under a rate code framework. This model provides an explanation for the tuning to peripheral ITDs reported in brainstem recordings of mammals (*McAlpine et al., 2001*; *Hancock and Delgutte, 2004*; *Pecka et al., 2008*). We followed the model's method of calculating Fisher information of individual neurons' IPD-tuning curves (*Figure 4C*, top) and their proposed distribution of best IPDs of the neuronal population in humans (*Figure 4C*, bottom-left) for computing the population's Fisher information of firing rate (*Figure 4C*, bottom-right). The population's Fisher information of firing rate across ITD and frequency, which

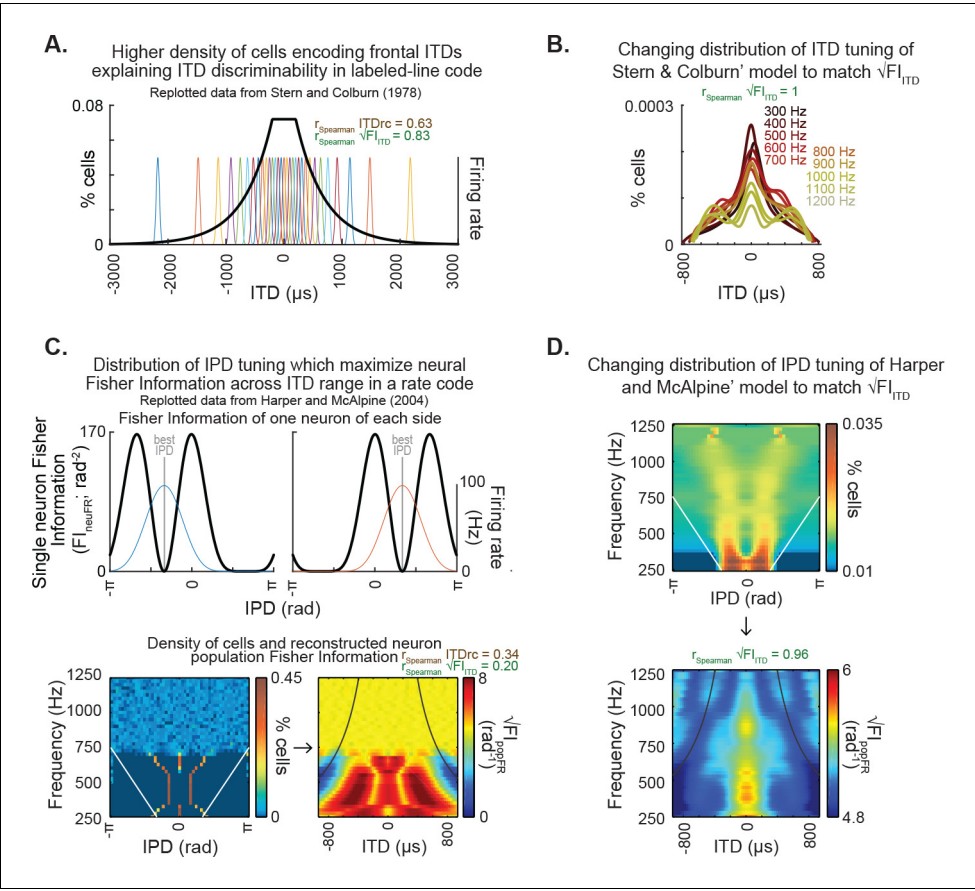

**Figure 4.** Classic models of neural properties underlying ITD discriminability and their potential for explaining encoding of ITD statistics. (A) Distribution of internal delays replotted from *Stern and Colburn, 1978*, *Figure 2b*, which proposes a higher density of pairs of fibers encoding frontal ITDs. (B) The density of pairs of fibers proposed by *Stern and Colburn, 1978* as the mechanism underlying ITD discriminability could effectively achieve the representation of ITD statistics: the density was adjusted to match the pattern of $\sqrt{FI}_{ITD}$. Note that the adjusted distribution largely preserves the shape of the distribution of the original model. (C) Distribution of IPD-tuning maximizing coding across the physiological range of ITD, as proposed by *Harper and McAlpine, 2004*. Top, single-neuron Fisher information as a function of IPD. Bottom-left, distribution of best IPDs brain across frequency expected for humans under the framework proposed by the authors; white straight lines indicate physiological range determined by the distance between ears. Bottom-right, reconstructed neuron population Fisher information, converted from IPD to ITD for each frequency for obtaining the predicted ITD discriminability; black curved lines indicate the π-limit, beyond which ITD cues become ambiguous within narrow frequency bands. Spearman correlation coefficients for the relationship between population Fisher information and ITDrc and $\sqrt{FI}_{ITD}$ outlined above. (D) The IPD-tuning distribution proposed by *Harper and McAlpine, 2004* as a mechanism underlying ITD discriminability was adjusted for matching the neuron population Fisher information to ITDfi. Top, the neuron distributions matching ITD statistics depict best IPDs away from midline across frequency, consistent with a coding strategy based on two clustered subpopulations tuned to IPDs away from the front (*McAlpine et al., 2001*; *Harper and McAlpine, 2004*; *Hancock and Delgutte, 2004*; *Pecka et al., 2008*). Bottom, the neuron population Fisher information highly correlated with the ITD statistics.

would correspond to the predicted ITD discriminability by this neural model, showed low correlation with both ITDrc and $\sqrt{FI}_{ITD}$ (*Figure 4C*, bottom-right). This is expected from *Harper and McAlpine, 2004* model's premise of a uniform maximization of Fisher information by firing rate across the physiological ITD range, as it differs from the higher frontal information predicted by the current study. The *Harper and McAlpine, 2004* model is, however, consistent with the low prediction accuracy of dITD thresholds and ITD deviance detection results (*Figures 2* and *3*).

Furthermore, we also tested whether a model relying on the firing rate Fisher information could also match ITD statistics (*Figure 4D*). Towards this goal, the density of neurons tuned to each IPD of the model was changed in order to make the neural information correlated with $\sqrt{FI_{ITD}}$. This consisted of designing neural populations with a density of preferred ITDs resulting in Fisher information of their ITD tuning curves being equal to $\sqrt{FI_{ITD}}$. The IPD-tuning distribution that generates a neural population Fisher information of firing rate matching $\sqrt{FI_{ITD}}$ differs from the IPD-tuning distribution originally proposed by *Harper and McAlpine, 2004*. However, matching ITD statistics under this coding framework predicted neurons' IPD-tuning clustered within peripheral IPDs across the frequency range tested, consistent with reports of brainstem recordings in mammals (*McAlpine et al., 2001*; *Hancock and Delgutte, 2004*; *Pecka et al., 2008*).

These results suggest the mechanisms underlying ITD discriminability proposed by *Stern and Colburn, 1978* and *Harper and McAlpine, 2004* are consistent with coding frameworks adapted to natural ITD statistics, providing a plausible biological connection between the coding of sensory statistics and perceptual functions.

## Discussion

Different explanations for the greater discriminability of sound locations in the frontal region have been proposed, including an uneven density of brainstem ITD-sensitive neurons under a labeled-line code framework (*Colburn, 1977*; *Stern and Colburn, 1978*) and greater change in the firing rate of these neurons as a function of azimuth in the front compared to the periphery under a rate code framework (*McAlpine et al., 2001*; *Harper and McAlpine, 2004*). Mechanisms based on the spatial information carried by auditory stimuli have also been invoked, such as the rate of change of ITD as a function of azimuth (ITDrc) (*Mills, 1972*; *Gelfand, 2016*). Our study proposes a new factor influencing the amount of spatial information carried by auditory stimuli, the ITDv. These statistics combined in the square root of ITD Fisher information ($\sqrt{FI_{ITD}}$) were good predictors of ITD discriminability and spatial novelty detection, supporting the hypothesis that natural ITD statistics determine the neural code underlying human sound localization. Finally, we showed that the models of *Stern and Colburn, 1978* and *Harper and McAlpine, 2004* can reflect the encoding of ITD statistics, thereby providing a functional connection between neural coding frameworks proposed by these models and experimental data on ITD perception.

Previous reports proposed connections between neural network properties and natural stimulus statistics by investigating the selectivity of midbrain neurons to the variability of spatial cues in the owl's auditory system (*Cazettes et al., 2014*; *Fischer and Peña, 2017*). These studies provided evidence of how sensory reliability could be represented (*Fischer and Peña, 2011*; *Rich et al., 2015*; *Cazettes et al., 2016*) and integrated into an adaptive behavioral command (*Cazettes et al., 2018*). Although properties of the neural mechanisms underlying human and owl sound localization differ in frequency range and putative ITD coding schemes (*Schnupp and Carr, 2009*), studies in both species support the concept that natural ITD statistics guide ITD processing.

This study specifically investigated whether ITDrc and ITDv, inherent in natural acoustic scenes, are relevant parameters determining ITD discriminability. We tested this hypothesis using discrimination thresholds obtained through free-field (*Mills, 1958*) and dichotic stimulation protocols that disabled natural ITDrc and ITDv statistics. We found that the integration of these ITD statistics based on Fisher information ($\sqrt{FI_{ITD}}$) was the best predictor of discrimination thresholds of spatial changes across frequency and location. This suggests that the neural code is adapted to the combination of ITDrc and ITDv statistics. However, the higher dichotic dITD thresholds for 250 Hz tonal stimuli (also reported by *Brughera et al., 2013*) constituted an outlier, indicating limited predictive power of ITD discriminability at this particular frequency by the ITD statistics investigated in this study. Although additional factors may determine discriminability in lower frequencies, our results overall are consistent with the notion that natural $\sqrt{FI_{ITD}}$ statistics can modulate the neural code underlying human sound localization.

We also implemented an MMN paradigm to obtain converging evidence that natural ITD statistics influence spatial perception. Although both ITD-change detection thresholds and the MMN novelty detection paradigms required discriminating a change in ITD, novelty detection also involves identification of a repetitive (standard) pattern. Furthermore, the combination of ITDrc and ITDv

($\sqrt{FI_{ITD}}$) was a better predictor of the deviance detection response, the MMN component, than ITDrc alone, which is also consistent with results of psychophysical discriminability thresholds.

*Parras et al., 2017* developed an approach designed to isolate the relative contribution of prediction errors and repetition suppression in novelty detection. In the present study, natural ITD statistics might have modulated novelty detection at both levels. However, the model that best described the MMN signals in the current study relied mostly on the ITD statistics of the standard stimulus for weighting the difference between the ITDs of standard and deviants stimuli primarily by the ITD statistics of the standards. Other factors with potential influence on detecting ITD changes in a reference location are attention and training. The novelty detection protocol controlled for these factors because the MMN indexes a brain response to detected deviations irrespective of attention or training. Our finding that prediction of novelty detection signals was based primarily on the ITD statistics of the standard stimulus is consistent with the interpretation that natural ITD statistics are critical for pattern detection. Our results indicate that when the standard stimulus is in a location of higher statistical discriminability, a 'stronger' standard is built, which makes deviance detection easier. We speculate that the different weights for standards and deviants are the result of the mechanism underlying the building up of a standard, which requires repetitive stimulation.

Finally, our results support classic neural models of ITD coding. The compatibility of ITD statistics with classic neural models of ITD coding suggest that ITD statistics provide a potential mechanism influencing the density distribution of ITD tuning. The critical parameter of the density distribution of fiber pairs encoding interaural delays that the *Stern and Colburn, 1978* model relied on to explain ITD discriminability was correlated with our ITD statistics. Thus, this model prediction matches the experimental data. Additionally, the coding scheme of the model proposed by *Harper and McAlpine, 2004* is a plausible framework for our results, in which ITD discriminability is predicted by the neural population Fisher information. When the neural population Fisher information was modeled to match the ITD Fisher information, the predicted distribution of ITD tuning resembled experimental observations in brainstem of mammalian species (*McAlpine et al., 2001*; *Harper and McAlpine, 2004*; *Hancock and Delgutte, 2004*; *Pecka et al., 2008*).

In sum, we found evidence that natural ITD statistics are correlated with auditory spatial perception, supporting the idea that these statistics may determine the density distribution of ITD tuning in the auditory system and influence auditory spatial perception. The consistency across subjects indicates that this information may be genetically encoded and conserved, and serve as a potentially adaptive evolutionary mechanism for approaching optimal performance. Such a mechanism would be useful where larger ITD changes are required for detecting shifts in location for regions of space and frequency levels at which ITD discriminability is naturally weaker. These results have clinical implications in identifying stimulus parameters that are relevant to spatial discrimination and novelty detection that may lead to the development of more efficient hearing-aid devices.

## Materials and methods

### HRTF measurement

The dataset used in this study consisted of head-related impulse responses collected at the Institute for Research and Coordination in Acoustics/Music (IRCAM) from 2002 to 2003, available to the public at the LISTEN HRTF website http://recherche.ircam.fr/equipes/salles/listen. The procedure was performed inside an anechoic room with walls that absorbed sound waves above 75 Hz. The pulse signals were played by TANNOY 600 speakers facing the subjects, at a distance of 1.95 m from the center of the head. Subjects were seated on an adjustable rotating chair with a position sensor placed on the top of the head, allowing recording only when the position was correct. The impulse sounds were collected with a pair of small Knowles FG3329 microphones, calibrated using a BandK 4149 microphone. These small microphones were mounted on a silicon structure, which occluded the ear canals of the subjects, avoiding resonance and placing the microphone sensor at the entrance of the ear canal. The signal captured by the microphones was driven to a custom-made amplifier with 40 dB gain, and recorded using an RME sound card interface with Max/MSP real time library which deconvolved the microphone signal.

## HRTF analysis

HRTF data from 51 subjects were included in the analysis (*Figure 1*). Head-related impulse responses (*h*) for the left (*L*) and right (*R*) ears corresponding to speaker locations at 0-degree in elevation and −90 to 90 degrees in azimuth (θ) were denoted as a function of time, $h_{L,θ}(t)$ and $h_{R,θ}(t)$. The azimuth in the database was sampled in 15-degree steps. Impulse responses were convolved with a white noise signal of 1 s duration *s(t)* to model the signals (*x*) received at the left and right ears:

$$x_L(t) = h_{L,θ}(t) * s(t)$$

$$x_R(t) = h_{R,θ}(t) * s(t)$$

where * denotes convolution.

This procedure transfers temporal and level effects of the filtering properties of the head to the convolved signal. These convolved signals were filtered by narrow-band filters modeling cochlear processing, using the gamma-tone filter bank from Malcolm Slaney's Auditory Toolbox, available in (https://engineering.purdue.edu/~malcolm/interval/1998-010). Gamma-tone filters are described in the following cochlear impulse response equation,

$$g(t;f_k) = t^3 e^{-\frac{t}{τ_k}} \cos(2πf_k t) U(t),$$

where *U(t)* is the unit step function and the center frequencies of the filters ($f_k$) ranged from 250 to 1250 Hz in 5 Hz steps. These center frequencies are within the range where ITD is a primary spatial binaural cue (*Rayleigh and Xii, 1907*) and also correspond with the frequency range of thresholds estimated by *Mills, 1958*. The time constants ($t_k$) were chosen such that the bandwidth of these filters matched the estimated bandwidth of human cochlear filters (*Glasberg and Moore, 1990*).

The outputs of the gamma-tone filter bank on the left ($y_L(t;f_k)$) and right ($y_R(t;f_k)$) sides were computed by convolving left- and right-ear input signals with gamma-tone filters,

$$y_L(t;f_k) = g(t;f_k) * x_L(t)$$

$$y_R(t;f_k) = g(t;f_k) * x_R(t).$$

Instantaneous phase was then computed for these output signals using the Signal Processing Toolbox (Mathworks). The instantaneous phase was computed as the phase (argument; *arg*) of the analytic representation of the signal,

$$θ_L(t;f_k) = arg\{y_L(t;f_k) + i\hat{y}_L(t;f_k)\}$$

$$θ_R(t;f_k) = arg\{y_R(t;f_k) + i\hat{y}_R(t;f_k)\}$$

where *y* is the signal and *ŷ* is its Hilbert transform.

For each azimuth (θ) and frequency range ($f_k$), we then calculated the instantaneous interaural phase difference (IPD) over time,

$$IPD(t;f_k) = θ_R(t;f_k) - θ_L(t;f_k)$$

where *IPD(t;f$_k$)* is in radians.

The circular mean and standard deviation of the instantaneous IPD over time was then computed. To avoid the ITD rate of change being corrupted by artificial values caused by phase ambiguity, we unwrapped (MATLAB function) the mean IPD over azimuth, and subtracted the value 2π repeatedly (from all IPD values jointly) until the IPDs corresponding to midline locations returned to the value before the shift. Finally, the circular mean and standard deviation of IPD was converted to ITD (in μs) using the following equation:

$$ITD = \frac{10^6 IPD}{2πf}.$$

All the HRTF analysis steps described above are shown in *Figure 1A*. The mean ITD across azimuth was interpolated using a cubic spline (*Figure 1B-left*), and the rate of change of ITD across azimuth (ITDrc) was calculated as the derivative of this curve. The standard deviation of ITD (ITDv) was

interpolated using the same method (*Figure 1B-right*) and the derivative of ITDv was calculated as the derivative of this curve.

We next combined the ITDrc and ITDv in a single quantity that is related to the discriminability of sound locations using ITD. The discriminability of a stimulus θ based on a measurement m(θ) is often described in terms of the Fisher information

$$FI(\theta) = -E\left[\frac{\partial^2}{\partial \theta^2}\log p(m|\theta)|\theta\right],$$

where p(m|θ) is the conditional probability of the measurement given the stimulus. In our analysis, θ refers to azimuth location, while m(θ) is the ITD computed from the output of the left and right cochlear filters at a given frequency which is used to infer the stimulus.

We also assume that the conditional probability of ITD given azimuth p(ITD|θ) is a Gaussian distribution with mean μ(θ) and standard deviation σ(θ). Substituting the Gaussian conditional probability p(ITD|θ) into the definition of the Fisher information (*Abbott and Dayan, 1999*), the formula reduces to

$$FI(\theta) = \left(\frac{\mu'(\theta)}{\sigma(\theta)}\right)^2 + 2\left(\frac{\sigma'(\theta)}{\sigma(\theta)}\right)^2,$$

where μ′(θ) is the ITD rate of change (ITDrc), σ(θ) is the standard deviation of ITD (ITDv) and σ′(θ) is the derivative of the standard deviation of ITD (ITDv′) with respect to azimuth. Discrimination thresholds have been shown to be proportional to the square root of the Fisher information (*Abbott and Dayan, 1999*), therefore we computed the square root of the ITD Fisher information ($\sqrt{FI}_{ITD}$).

The second term in the equation, which is often absent in calculations of Fisher information, is included in our analysis because the standard deviation of ITD changes with direction. Note that when the derivative of standard deviation σ′(θ) is zero, the square root of the Fisher information simplifies to μ′(θ)/σ(θ), the same as ITDrc/ITDv. This first term in the equation is conceptually similar to the d-prime metric; however, while d-prime is the subtraction of two means divided by the standard deviation, this part of the equation is the derivative of the mean divided by the standard deviation.

Finally, azimuth was converted to ITD (using the relationship between azimuth vs. ITD determined from the HRTFs), obtaining an estimate of the ITD statistics across frequency and ITD. The ITD statistics were computed for each subject, then the median and interquartile range of them was computed for each combination of azimuth and frequency across subjects (*Figure 1C*).

## Estimation of spatial discriminability thresholds from previously published datasets (*Mills, 1958*)

Human spatial discriminability thresholds were estimated in the classic *Mills, 1958* study. Data collection for this study was performed inside an anechoic room with a movable speaker, which delivered tones of different frequencies. The three participants were blindfolded and had their heads fixed by a clamp mounted on the chair on which they were sitting. In each trial, a 1 s duration 'reference azimuth' tone was played first, and 1 s after, the same signal played again after the speaker was moved slightly to the left or to the right. Subjects reported the left or right change using an interface box. Psychometric functions were obtained plotting the proportion of judgments to the left and to the right against the angle between reference and test locations. The azimuth changes leading to 75% responses to the right and to the left were estimated by linear interpolation. The threshold angle for discriminating a change was estimated by dividing the distance between these values by 2.

To convert azimuth to ITD, *Mills, 1958* used binaural microphones placed inside a dummy-head ears. The ITD corresponding to the reference azimuth and the IPD corresponding to the threshold azimuth change were measured visually from signal traces using an oscilloscope. Threshold IPDs vs. reference ITDs were plotted in a logarithmic Y-axis vs. linear X-axis scale and linear curves were fit to the data. For the current study, we extracted data points of threshold dIPD across reference ITD and frequency from *Mills, 1958*. Threshold dIPD was converted to threshold dITD (using the same equation described in the HRTF analysis section).

## Estimation of spatial ITD discriminability thresholds

A test was designed to estimate detection thresholds of changes in stimulus ITD (dITD) across specific frequencies of interest for this study. Healthy adult subjects were included in the sample (N = 24; 12 females and 12 males; mean age 28.0 ± 8.6; five left-handed and 19 right-handed; 19 from São Paulo and five from New York). After the procedure was described to the subjects, they provided written informed consent. The protocol was approved by the Ethics Committee of Universidade Federal do ABC and by the Internal Review Board of the Albert Einstein College of Medicine, where the study was conducted. There were no distinct groups in the experiment. All subjects had no reported history of neurological disorders or hearing impairments.

Pilot measurements of dITD thresholds were initially conducted in 10 subjects, using the same combination of frequencies and ITDs across subjects. This pilot experiment, which lasted approximately 150 min, was performed in up to five sessions per subject. Results from these measurements already showed that the ratio between ITDrc and ITDv leads to good prediction for frequencies above 250 Hz. Based on feedback from subjects undergoing pilot measurements, a shorter protocol lasting about 60 min was designed, which was conducted in 24 subjects, leading to the reported dITD thresholds results.

In this computer-based test, subjects were presented with 65 dB (A scale) tones within a range of frequencies through headphones calibrated with an Instrutherm DEC-460 decibel meter or a B&K 4947 microphone with an artificial ear. Trials started by pressing the spacebar key. In each trial, two binaural tones were presented in sequence. The ITD of both sounds started at the same value (reference ITD) and changed by different amounts (dITD) in the second half of either the first or the second sound in the sequence. Subjects were instructed to press the keys '1' or '2' depending on which sound in the pair they perceived a shift in location, and press a given key twice if confident, or alternate both (1 and 2) keys if unable to perceive a shift or unsure about it. Trials could be repeated as many times as needed by pressing the spacebar key. Feedback sounds indicated whether each of the pressed keys was correct or wrong.

The range of reference ITDs spanned from frontal (0 µs) to peripheral locations within the unambiguous range of ITD for each frequency. ITD change (dITD) varied from 1 µs to 200 µs towards the periphery to cover a range from unequivocally detectable and undetectable changes for each frequency. The direction of dITDs relative to the reference ITD was always away from the front, to avoid direction dependent biases affecting threshold measurements. Each condition (a given combination of frequency and reference ITD) was presented 29 times: four training trials with dITD 200 µs which were not computed and 25 testing trials which were used for estimating the dITD threshold. An initial training block used 750 Hz tones and −50 µs reference ITD for all subjects and was not included in the analysis. The following 20 blocks presented tones of frequencies 250, 500, 750, 1000 or 1250 Hz, and reference ITDs from −500 to 0 varied in steps of 100 µs. Reference IPDs of absolute values smaller than π/2 radians were included, to avoid phase ambiguity. The sequence of conditions was randomly varied across subjects.

Ongoing estimation of dITD thresholds was conducted from trial 1 of each block, to optimize the test estimate. A psychometric sigmoid curve was progressively fit to a plot of correct responses (assigned 1) and incorrect or unsure responses (assigned 0) as a function of dITD; the dITD corresponding to 0.5 accuracy was selected as the estimated dITD threshold. Preliminary dITD thresholds were computed from subsets of trials within varying dITD ranges. The first six trials ranged from dITDs 10 to 190 µs, spaced by 36 µs. From these trials, a first preliminary dITD threshold was estimated. In the following six trials, dITD was varied from −50 to 50 µs in steps of 20 µs, centered on the first preliminary dITD threshold; by the end of trial 12 a second preliminary dITD threshold was estimated using trials 1 to 12 using the same sigmoid fitting procedure. A third preliminary dITD threshold was then estimated using trials 1–18. The final seven trials ranged from −21 to 21 µs spaced by 7 µs, centered on the third preliminary dITD threshold. When a set of dITDs centered on a preliminary dITD threshold extended beyond the 1 to 200 µs range, dITDs were adjusted to fall within this range. While this procedure permitted an efficient estimate of dITD thresholds, it did not yield plausible dITD threshold estimates in cases where subjects provided largely random responses across dITDs. To address this limitation, a nonparametric receiver operating characteristic (ROC) classifier was conducted offline to independently verify the validity of estimated dITD thresholds.

For estimating threshold dITDs, an ROC classifier was computed over the 25 trials of each condition for each subject. The threshold was estimated by averaging the subset of possible thresholds within the 1 to 200 μs range that jointly maximized the number of correct (hit) responses and minimized false positive (type II error) ones. This optimization was obtained by selecting candidate dITD thresholds within a minimum euclidean distance from the perfect discrimination (i.e. 100% hit rate and 0% type II error rate) yielded by the ROC analysis. The ROC classifier was robust enough to estimate consistent dITD thresholds to all conditions from all subjects (*Figure 2C-left*). The threshold within 1–200 μs range estimated by the sigmoid fitting method were significantly correlated with those estimated by the ROC classifier ($r_{Pearson}$ = 0.85).

## Prediction of spatial discriminability thresholds by ITD statistics

An initial analysis of the ranked (Spearman's) correlation coefficients was performed for the relationship between the threshold dITD averaged across subjects and the ITD statistics of the reference ITD (middle and right plots of *Figure 2C*). Spearman's correlations, which were computed from averages of dITD thresholds over subjects across multiple conditions (*N* number of combinations of reference ITD and frequency), were used to assess the monotonicity of this relationship. Since the *N* in this analysis reflects the number of conditions, not the number of subjects, the standard statistical power analysis does not apply. Accordingly, p-values were not computed for this correlation analysis.

Additional analysis for the selection of dITD thresholds' best predictors was performed using linear mixed-effect models (LMM; *Magezi, 2015*), classifying ITD statistics across stimulus conditions as 'fixed factor' and participants as 'random factor'. LMM analysis assumes linearity between measures and predictors, then we inspected whether the relationship between dITD thresholds and each of the predictors was linear. Although relationships were mostly linear, some were best fitted by a power function. In these cases, we applied the standard method for achieving linearity by log-transforming both predictors and dITD thresholds. Linear regressions of the relationship between the multiple log-transformed dITD thresholds collected from each subject and log-transformed ITD statistics were performed, and Akaike Information Criterion (AIC) computed. The AIC analysis was used to compare the performance of each model, relying on both the number of model parameters and sample size (i.e. number of subjects) as a metric of goodness of fit; the lowest AIC corresponds to the best model. Since AIC is a relative quantity (i.e. the actual value brings no information alone), we normalized the AIC values by subtracting the minimum AIC observed for that behavioral measure (dAIC), which corresponds to the information loss compared to the best available model. Models with dAIC between 0 and 2 were considered equally good in their prediction capacity. Sample sizes were made several times higher than the number of parameters of our models to ensure samples were sufficiently large.

## Collection and analysis of the mismatch negativity (MMN) component

Healthy adult participants were included in the sample (N = 33, 16 females; 17 males; mean age 29.5 ± 4.8; all right-handed). After the procedure was described to the subjects, they provided written informed consent. The protocol was approved by the Institutional Review Board of the Albert Einstein College of Medicine, where the study was conducted. All subjects had no reported history of neurological disorders and passed a bilateral hearing screening (20 dB HL or better at 500, 1000, 2000, and 4000 Hz).

A statistical power analysis for the MMN component using a stringent minimum MMN amplitude of −0.5 μV (SD 0.7 μV) revealed substantial power (1-β=0.87) with an alpha level of 0.05 in 30 adult subjects. We performed the first set of conditions (1–10) in 17 subjects and found that adjusting the difference between ITD of standard and deviant by a weighted average of the ratio between ITDrc and ITDv of the stimuli ITD; 16 additional subjects were recruited for a second set of conditions (11–20), replicating the initial findings. No significant difference was found between groups and therefore the analysis reported in the manuscript was performed on the pooled data of all 33 participants. Participants were seated inside a sound-attenuated booth (IAC Acoustics, Bronx, NY) in front of a TV screen where a subtitled muted movie was presented. Sound was delivered by a Neuroscan StimAudio system through insert earphones (3M Eartone 3A) calibrated to 53 dB (A-weighted) using a B

and K 4947 microphone with an artificial ear. Sound signals were 50 ms duration tones (25 ms rise-fall time) of different frequencies and ITDs synthesized with MATLAB software.

Participants listened to oddball sequences presenting repetitive ('standard') tones embedded with sporadic ('deviant', 15% of the trials) tones with a 300 ms inter-stimulus interval (*Figure 3A-left*). Each subject was presented with 10 conditions, which differed in frequency and ITD. Subjects 1 to 17 performed the following conditions ('ITD standard' vs. 'ITD deviant' at 'tone frequency'): (1) −590 vs. −295 μs at 400 Hz; (2) −295 vs. 0 μs at 400 Hz; (3) 0 vs. −295 μs at 400 Hz; (4) −295 vs. 295 μs at 400 Hz; (5) 295 vs. −295 μs at 400 Hz; (6) −295 vs. 590 μs at 400 Hz; (7) 590 vs. −295 μs at 400 Hz; (8) −590 vs. −295 μs at 600 Hz; (9) −295 vs. 0 μs at 600 Hz; and (10) 0 vs. −295 μs at 600 Hz. Subjects 18 to 33 performed the conditions (11) 0 vs. −499 μs at 400 Hz; (12) −499 vs. 0 μs at 400 Hz; (13) 0 vs. −159 μs at 400 Hz; (14) −159 vs. 0 μs at 400 Hz; (15) 0 vs. −499 μs at 550 Hz; (16) −499 vs. 0 μs at 550 Hz; (17) 0 vs. −499 μs at 650 Hz; (18) −499 vs. 0 μs at 650 Hz; (19) 0 vs. −159 μs at 650 Hz; and (20) −159 vs. 0 μs at 650 Hz. Each condition was presented in 3 blocks of 474 trials; the block order was randomized for each subject. Each of the conditions was presented three times during the experimental session; the order of blocks was randomized for each subject. The first trials of each block (18 standards + one deviant) were used for training and not included in the analysis. Results reported included the following trials (385 standards and 70 deviants) for each subject. MMN values for each condition were estimated by subtracting mean ERP signals of 210 deviant trials by the mean of 1155 standard trials; after removal of trials with artifacts (see below). Sessions lasted approximately 2.5 hr, including placement of electrode caps and breaks during the EEG recording.

EEG was recorded with Neuroscan SynAmps and a 32-channel electrode cap following the modified international 10–20 System, including electrodes on the nose (reference), P09 (ground) and left and right mastoids (LM and RM, used for offline analysis). A bipolar configuration was used between an external electrode below the left eye and the FP1-electrode position for measuring vertical electro-oculogram (EOG). The signal was recorded at 500 Hz sampling rate using a band-pass from 0.05 to 100 Hz. Impedances were maintained below 5 kOhms.

To measure the MMN, EEG signals from each subject were processed as follows: (1) 20 Hz low-pass filtering; (2) pooling (by concatenating) all EEG signals obtained during sound stimulation; (3) removal of eye-blink and saccade artifacts by performing Independent Component Analysis and reconstructing the signal without components correlated to the EOG; (4) selection of 600-millisecond epochs around sound presentation (−100 to 500 ms from sound onset); (5) removal of linear portions of EEG signals and exclusion of trials containing voltage deflections larger than 75 mV; (6) re-reference of the EEG recordings to the mastoid recordings by subtracting the average mastoid signal from the other channels; (7) ERP measurement, averaging separately signals evoked by standard and deviant trials (the first 19 trials were used for subject training and excluded from the analysis); (8) subtraction of standard from the deviant ERPs to compute the MMN; (9) identification of time bin containing the MMN peak in signals from the FZ channel (frontal EEG electrode, in the middle of the forehead) averaged across subjects, for each condition; (10) measurement of MMN peak for each subject and condition within this time bin.

Grand-averages of ERPs recorded at FZ electrodes were computed for standard and deviant trials across all subjects and conditions; for estimating the MMN topography, the signal from each electrode in the time bin of the peak MMN was averaged across subjects and conditions (*Figure 3A-right*).

## Prediction of MMN by ITD statistics

An initial analysis of the ranked (Spearman) correlation coefficients was performed for the relationship between the MMN peak amplitude averaged across subjects and the absolute ITD difference between standard and deviant stimuli, multiplied by the weighted sum of ITD statistics estimated for both standard and deviant stimuli (*Figure 3C-left*). Additional LMM analysis (described in "Prediction of spatial discriminability thresholds by ITD statistics'' section) was used to compare performance across predictors of MMN peak amplitude. LMM analysis was conducted on MMN peak as the dependent variable, absolute ITD difference multiplied by the weighted sum of ITD statistics as 'fixed' factor and participant as 'random' factor. Since the relationship between MMN data and predictors followed a power function, it was linearized using log transformation in both measures. No

outliers were detected or excluded. The AIC method was used for comparing the models (described in "Prediction of spatial discriminability thresholds by ITD statistics'' section).

## Neural models

Two seminal models (*Stern and Colburn, 1978*; *Harper and McAlpine, 2004*) addressing discriminability of azimuth positions in acoustic space based on ITD were used to examine the potential link between the brain representation of sensory statistics and perceptual functions.

The relative number of fiber pairs encoding interaural delays, p(τ), used as predictors of ITD discriminability (*Figure 4A*), was extracted from *Stern and Colburn, 1978*. To test whether ITD statistics could be represented by this model, the p(τ) parameter was adjusted to match statistics, by normalizing ITD statistics from 0 to 1 and scaling the resulting data to obtain a probability distribution where the sum of all probabilities was equal to 1 (*Figure 4B*).

Fisher information from single-neuron IPD-tuning curves and the optimal population distribution of IPD-tuning estimated for humans were extracted from data reported in *Harper and McAlpine, 2004*. The M-shaped Fisher information curve was positioned at the best IPD of each neuron within the distribution, to obtain an estimate of the population Fisher information across IPD and frequency. The neuron population Fisher information across IPD was converted to ITD, obtaining the prediction of ITD discriminability induced by the neuron distribution (*Figure 4C*). To test whether ITD statistics could be represented by parameters of this model, for each frequency, we generated two midline-mirrored gaussian neural population distributions with random mean IPD tunings and standard deviations from 0 to π, then selected the distribution that displayed the highest Pearson correlation between Fisher information and ITD statistics, then constant were summed and multiplied to the density values in order to to obtain one in slope and intercept of a linear fit. Finally, the density across all frequencies were corrected to generate probability one.

All data processing was performed in MATLAB (Mathworks) using built-in or custom-made routines. The datasets generated and analyzed in the current study are available in https://doi.org/10.5061/dryad.h70rxwdf9.

## Acknowledgements

This work was supported by the NIH BRAIN Initiative (Grant NS104911) and by the NIDCD (Grants DC004263 and DC007690). We thank André Cravo, Boris Marin, Fanny Cazettes, Gervasio Batista, Michael Beckert, Peter Claessens and Ruben Coen Cagli for useful discussions and comments on the manuscript, and Renee Symonds, Kelin Brace and Huizhen (Joann) Tang for support with troubleshooting during early stages of data collection. We also thank eLife journal editors and reviewers for their thoughtful and constructive comments.

## Additional information

### Funding

| Funder | Grant reference number | Author |
| --- | --- | --- |
| National Institutes of Health | NS104911 | José L Peña<br>Brian J Fischer |
| National Institute on Deafness and Other Communication Disorders | DC004263 | Elyse S Sussman |
| National Institute on Deafness and Other Communication Disorders | DC007690 | José L Peña |

The funders had no role in study design, data collection and interpretation, or the decision to submit the work for publication.

## Author contributions
Rodrigo Pavão, Conceptualization, Resources, Data curation, Software, Formal analysis, Validation, Investigation, Visualization, Methodology, Writing - original draft, Writing - review and editing; Elyse S Sussman, Conceptualization, Resources, Data curation, Supervision, Funding acquisition, Methodology, Project administration, Writing - review and editing; Brian J Fischer, Conceptualization, Software, Funding acquisition, Methodology, Writing - review and editing; José L Peña, Conceptualization, Resources, Data curation, Supervision, Funding acquisition, Investigation, Writing - original draft, Project administration, Writing - review and editing

## Author ORCIDs
Rodrigo Pavão (iD) https://orcid.org/0000-0002-6857-8963
Elyse S Sussman (iD) https://orcid.org/0000-0002-1013-0621
Brian J Fischer (iD) http://orcid.org/0000-0001-5786-0544
José L Peña (iD) https://orcid.org/0000-0001-6773-5640

## Ethics
Human subjects: This study was performed in accordance with the NIH Human Subjects Policies and Guidance and with the Brazilian National Health Council, and it was approved by the Internal Review Board of the Albert Einstein College of Medicine (#1999-023) and Ethics Committee of Universidade Federal do ABC (#2968291).

## Decision letter and Author response
Decision letter https://doi.org/10.7554/eLife.51927.sa1
Author response https://doi.org/10.7554/eLife.51927.sa2

# Additional files
## Supplementary files
• Transparent reporting form

## Data availability
All data generated or analysed during this study are included in the manuscript and supporting files.

The following dataset was generated:

| Author(s) | Year | Dataset title | Dataset URL | Database and Identifier |
|---|---|---|---|---|
| Pavão R, Sussman ES, Fischer BJ, Pena JL | 2019 | Anticipated ITD statistics are built into human sound localization | https://doi.org/10.5061/dryad.h70rxwdf9 | Dryad Digital Repository, 10.5061/dryad.h70rxwdf9 |

The following previously published datasets were used:

| Author(s) | Year | Dataset title | Dataset URL | Database and Identifier |
|---|---|---|---|---|
| IRCAM | 2003 | LISTEN HRTF DATABASE | http://recherche.ircam.fr/equipes/salles/listen/ | IRCAM LISTEN HRTF DATABASE, listen |

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
