## [Decision Letter]

**Acceptance summary:**

The natural statistics of spatial cues vary with sound source location, and this study supports the hypothesis that these statistics are represented in the neural code for interaural time differences and influence spatial perception.

**Decision letter after peer review:**

Thank you for submitting your article "Anticipated ITD statistics are built into human sound localization" for consideration by *eLife*. Your article has been reviewed by three peer reviewers, including Catherine Emily Carr as the Reviewing Editor and Reviewer #1, and the evaluation has been overseen by Andrew King as the Senior Editor. The following individuals involved in review of your submission have agreed to reveal their identity: Michael Pecka (Reviewer #3).

The reviewers have discussed the reviews with one another and the Reviewing Editor has drafted this decision to help you prepare a revised submission.

Summary:

This manuscript investigates whether the human brain exploits invariant spatial statistics to optimize sound source location discriminability. The authors first establish the presence of subject-invariant interaural time difference (ITD) statistics. After identifying potential physiological mechanisms that would render the neural ITD processing sensitive to these statistics, they then demonstrate the ability of those ITD statistics to predict human discrimination thresholds. Finally, an oddball sequence paradigm and EEG measurements are used to test to what extent ITD statistics processing can be detected in the human brain. The statistical characterization that is presented is compelling and interesting. However, several major concerns were raised that will have to be addressed satisfactorily for the paper to be deemed acceptable for publication. The presentation is opaque, and we had difficulty understanding how the authors computed ITDv. We were also not convinced that the data presented can distinguish whether ITDv and ITDrc simply influence ITD processing or whether the brain anticipates the location-specific invariance of these statistics and exploits them to optimize ITD discrimination.

Essential revisions:

1) The title and main conclusion are an overstatement. To justify these statements (anticipation etc), the authors would have to show that "unnatural" ITDv (i.e. scrambled higher-order stats) deteriorate ITD discrimination. Alternatively, the statements should be modified / toned down to more accurately reflect the significance of the findings.

2) It is not clear how the authors computed either the ITD rate of change with azimuth (ITDrc) or ITD variability (ITDv) (they mention across-frequency interference that changes over time). Perhaps an example and equations in the methods would make this clearer.

The critical contribution made by this paper is the introduction of ITD_v. The data for ITD_{rc} and ITD_v come from the LISTEN data base measured at IRCAM long ago. This database consists of physical measurements, interaural differences as a function of sound source location in free field for more than 50 individual heads. Unfortunately, it is not clear how ITD_v is determined. It is said to come from a variability in head related transfer functions (HRTFs) over time. But HRTFs do not normally depend on time. HRTFs describe wave filtering from a source in space to some chosen point in the ear. Why should they depend on time? Does ITD_v come from some estimate of the motion of listeners during the action of sound localization? Does it represent variability seen at IRCAM from one experimental session to the next for a given listener? The basis for the calculations presented here is a total mystery.

The critical variable ITD_v is shown in Figure 1D. There, the variability appears to be small for small ITD values (source near the geometrical midline) and it appears to be large for large ITD values (source out to the side). However, if the temporal variation of ITD is caused by inadvertent listener rotation, then the sensitivity pattern ought to be the opposite of what appears in Figure 1D. This is so because the sensitivity of ITD to azimuth is steepest near the midline and becomes quite flat at large source azimuths.

3) The manuscript is very hard to follow. The reasoning is obscure, with mathematical operations described in words instead of equations. Thus, the substance of the paper hides within a forest of details. We counted more than 120 figures. This is too many.

4) The evidence for some of the central claims of the study is either not presented sufficiently clear or tested only indirectly. We understand that the manuscript addresses an important question, and we understand the need to explain many rather complicated physical details and methodological constraints. One reviewer commended the authors for their ability to summarize and explain their data in a manner that is accessible to a broader readership. Nevertheless, the "anticipation of characteristic ITDv / statistics built into the brain" argument is not directly shown by the data that the authors provided (see next point).

5) We are not convinced that data that are presented can distinguish whether ITDv and ITDrc simply influence ITD processing or whether the brain anticipates the location-specific invariance of these statistics and exploits them to optimize ITD discrimination. I.e. the reported increase in variability at lateralized positions will certainly partially account for deterioration of JNDs (as will the decrease in ITDrc), and a large amount of data in support of this notion is presented.

Does changing higher-order statistics indeed deteriorate ITD JNDs (or dITD)? The experiment summarized in Figure 4 touches on this idea (loss of ITDv due to the use of in-ear earphones results in elevated thresholds), however, as the authors noted, this threshold shift could also be (partially) due to the use of naïve listeners or other factors and thus is inconclusive. Likewise, the MMN paradigm (Figure 5) is unconvincing in this respect as well. The authors first state that the paradigm design avoids the presence of ITDv. They then show that the recorded MMN data data is best explained by a model including D-stat (i.e. ITDrc and ITDv) and argue that this finding is evidence for the anticipation of these statistics. While we agree that this finding is suggesting potentially "anticipatory processing", it is very difficult to judge the significance of the model performances without a control condition (including naturalistic free-field statistics). Some sort of comparative analysis (i.e. the manipulation of the higher-order statistics) to directly show whether neuronal processing is adapted to a specific statistical distribution would be required to justify the strong statements made in the Title, Abstract, and Introduction (last paragraph) about anticipation.

[Editors' note: further revisions were suggested prior to acceptance, as described below.]

Thank you for submitting your article "Natural ITD statistics are built into human sound localization" for consideration by *eLife*. Your article has been reviewed by three peer reviewers, including Catherine Emily Carr as the Reviewing Editor and Reviewer #1, and the evaluation has been overseen by Andrew King as the Senior Editor. The following individuals involved in review of your submission have agreed to reveal their identity: Michael Pecka (Reviewer #3).

The reviewers have discussed the reviews with one another and the Reviewing Editor has drafted this decision to help you revise the manuscript. Please note, however, that we wish to see a point-by-point response to the main comments set out below, as this will determine whether we will be willing to consider another revision. We have summarized our extensive discussion in the following, and we have also appended the original (pre-consensus) reviews, for your edification.

Our expectation is that the authors will eventually carry out the additional experiments and report on how they affect the relevant conclusions either in a preprint on bioRxiv or medRxiv, or if appropriate, as a Research Advance in *eLife*, either of which would be linked to the original paper. This would be relevant if you wished to shorten the paper, and publish some experiments in a second publication.

Essential revisions:

1) We all regard the use of ITD statistics, specifically, the derivative (rate of change) of the mean ITD over azimuth (ITDrc) and the standard deviation of ITD (ITDv) over time as a major strength of the paper. The finding that the Stern and Colburn, (1978) and Harper and McAlpine, (2004) models can reflect a representation of ITD statistics provides a strong connection between the neural coding and psychophysics. Thus, neural density may reflect the Fisher Information for azimuthal angle changes, and the paper could be motivated by explaining why the ITD sensitivity is not equally good within the entire physiological relevant ITD range.

2) Our major concern is the wording, which often seems vague. To excerpt from a review "At the moment, the manuscript reads as if the ITD statistic is actually a detection cue, like "Oh, I now perceive the variance! The sound must come from this direction." We think you mean that fewer neurons code sound directions where there is less Fisher information (large ITDs, typically generated from large azimuthal angles) than where there is more information from the ears (small ITDs from mid-line directions). If this is what you mean, please try to be clearer and avoid phrases like terms "statistic is built in" and "the brain anticipates".

We have extensively discussed what you mean with respect to ITD sensitivity. You imply that ITD sensitivity (and the underlying neural processing) is improved by the naturally occurring ITDv, but you don't explain how this might work, i.e. you show that the brain uses the additional variance information for ITD calculation, but whether this information necessarily follows natural statistics is not clear. Furthermore, the wording of "anticipation" implies that the brain has learned what statistics exactly define naturalistic ITDv, which is not reflected in your data. Our uncertainty about what you mean motivates point 3 below.

A third point of confusion relates to D-stat. This has the highest correlation with the Mills data, and the way we understand (or interpret) your results and discussion is that even though no ITDv is present in the stimuli, the perception is explained by neural processing that incorporates the ITDv that would normally be present for these stimuli. And since the ITDv is not present, the brain must "know" (i.e. have learned) what these stats are. Please clarify if this is what you mean.

3) As you can tell from the above questions, our major concern is that the writing is still unclear. We have spent two weeks discussing what we think you mean. Therefore, we believe that there is potential to further clarify the wording and explanations. For example, with respect to understanding what is meant by "Natural ITD statistics are built into [the brain])", we would like to clarify whether you agree or disagree with the interpretation that "neural density may reflect the Fisher Information for azimuthal angle changes, etc" before you start revising the manuscript for a third time. Please clarify this along with your responses to the other points in a reply letter. Once we understand what you mean, we could suggest how/if to streamline the manuscript.

4) The reviewers have discussed a recommendation to omit some of your experiments, on the grounds that the additional information doesn't substantially add to the main message (see review #4), but rather raises questions that potentially distract from it. We would prefer to resolve the issues in Points 1-3 first.

Reviewer #1:

This is my second review of this interesting paper. The basic idea is very compelling, that natural ITD statistics are built into the neural circuits that mediate sound localization. I also like the combination of modeling and psychophysics.

Nevertheless, the presentation is quite complex. The authors have written the paper three times. Once, when first sent to the journal, prior to review, and next after the prior reviews. This version is much clearer and has less than half the number of figures.

Strengths – the use of ITD statistics, specifically, the derivative (rate of change) of the mean ITD over azimuth (ITDrc) and the standard deviation of ITD (ITDv) over time are both important insights. The authors' finding, that the Stern and Colburn, (1978) and Harper and McAlpine, (2004) models can implement a representation of ITD statistics, provided a strong connection between the neural coding and psychophysics. I feel the field has been arguing about this for decades, and that this represents a major insight.

Reviewer #3:

The authors substantially revised the manuscript by adding more detailed explanations for the calculations they made. They also employ more measured wording throughout the paper, now. I remain a bit skeptical about the suitability of their use of the terms "built in" and "anticipated" with regard to the conclusions of their findings.

Reviewer #4:

I was initially in full agreement with the critique made in the first review (reviewer #2 was not available for re-review) and did not understand the rebuttal by the authors. But it struck me that the authors are by no means new to this research area so that the apparent absence of sense in the manuscript can only be explained by a gross misunderstanding. It took a couple of days until it clicked, and I believe I understand now what the authors are trying to convey. My take on their research question is this:

Why is ITD discrimination not equally good within the entire ecologically relevant range but is better for midline directions? Because Mills' data are already converted into milliseconds of ITD, it cannot just be the geometry (e.g Kuhn, 1977). The authors argue: It is because in real life, with broad band stimulation, the interaural correlation arriving from the left and right ear at the coincidence-detecting neurons is reduced due to differing HRTF filtering. IPD and ILD fluctuations are also introduced if ITD is not perfectly compensated by internal delay line differences (e.g. differences in left and right axonal travel times). Thus, they say, why wasting large number ITD sensitive neurons to reduce neural noise on spatial locations, where spatial cues are degraded anyway (ITDv)? In addition, ITD changes per angular location change (ITDrc) are lower at these off-centre directions relative to midline directions. In other words, one could argue that the Fisher information (change of mean divided by variance – they call it D-stat) for angular changes in source direction from these direction is low and so requires less number of neurons to be coded than the far higher information from frontal directions. The so predicted density distribution of binaural detectors as a function of their best ITD can be validated with ITD discrimination experiments using single tones, as these do not get ITD (and ILD) fluctuations by going through HRTFs, auditory filters and delay lines. Because their D-stat derived with broadband stimuli fit Mill's data very well, but D-stat can itself actually not explain Mills ITD discrimination thresholds because they were obtained with pure tones and are expressed in millisecond (not azimuthal angle), one might conclude that whatever limits ITD discrimination with tones has been influenced by D-stat by experiencing everyday broadband sounds. I assume the authors imply that it is neural noise that became somehow adjusted to match D-stat. Neural noise per ITD is determined by the number of ITD coding neuron coding this ITD (e.g. Stern and Colburn, 1978; Harper and McApline, 2004).

In the case that this indeed is, what the authors intend to say, I believe their idea is well worth disseminating. I agree with the first review that the current presentation is opaque and the description vague and misleading. It is lacking logical line of thoughts that tells the reader this story. The choice of wording itself is confusing and become more precise and descriptive in order to convey the message. The narrative can and should be kept simple and specific as it is actually a straightforward story, which is certainly novel and worth exploring further.

I would like to give further advice: The basic idea is in my opinion sufficient for an initial publication and I would not "dilute" it with unnecessary details. This story does not need further experimental data than those well-accepted ITD threshold data by Mills, (1958). They are sufficient for the purpose. Showing own, partly contradicting experimental data, just adds confusion and distracts the reader from the main message. Similarly, the EEG data are irrelevant for the narrative and just lengthen the manuscript unnecessarily.

I agree that one can also base the neural density on the distribution of ITD tuning slope curves, assuming neurons are most sensitive to changes in ITD here (Harper and McAlpine, 2004). This is indeed worth mentioning because this thinking is becoming the popular. But the fact that an ITD tuning curve has two slopes just complicates the matter. For the sake of keeping the argument simple, I recommend not go into detail with the slope model, but illustrate the basic idea using the classic Jeffress-based model by Stern and Colburn, (1978).

---

## [Author Response]

Summary:This manuscript investigates whether the human brain exploits invariant spatial statistics to optimize sound source location discriminability. The authors first establish the presence of subject-invariant interaural time difference (ITD) statistics. After identifying potential physiological mechanisms that would render the neural ITD processing sensitive to these statistics, they then demonstrate the ability of those ITD statistics to predict human discrimination thresholds. Finally, an oddball sequence paradigm and EEG measurements are used to test to what extent ITD statistics processing can be detected in the human brain. The statistical characterization that is presented is compelling and interesting. However, several major concerns were raised that will have to be addressed satisfactorily for the paper to be deemed acceptable for publication. The presentation is opaque, and we had difficulty understanding how the authors computed ITDv. We were also not convinced that the data presented can distinguish whether ITDv and ITDrc simply influence ITD processing or whether the brain anticipates the location-specific invariance of these statistics and exploits them to optimize ITD discrimination.

We have revised the manuscript focusing on the clarification of general concepts and the methods to compute ITDv and analyze the data; we also reduced the number of figures. As mentioned in the summary above, the main take-home message of this work is that the brain anticipates location-specific ITD statistics invariant across contexts, and exploits them to optimize ITD discrimination. We assume that ITDv and ITDrc statistics influence ITD processing, as postulated in previous studies. However, the focus of our study was not to address the direct effect of ongoing or recent statistics on ITD processing. We believe that the claim suggested above, “The brain anticipates the frequency- and location-specific invariance of these statistics [ITDv and ITDrc across contexts] and exploits them to optimize ITD discrimination”, is a valid and accurate description, and added this statement to the abstract and conclusions.

Essential revisions:1) The title and main conclusion are an overstatement. To justify these statements (anticipation etc), the authors would have to show that "unnatural" ITDv (i.e. scrambled higher-order stats) deteriorate ITD discrimination. Alternatively, the statements should be modified / toned down to more accurately reflect the significance of the findings.

A way of testing if changes in ITDv affect ITD discrimination could be by manipulating interaural correlation. It is already known that decreasing interaural correlation deteriorates sound localization in humans. Previous work by co-authors of this manuscript has shown that behavioral biases observed in barn owls when interaural correlation is manipulated is consistent with a representation of ITDv, but whether this is also observed in humans has not been tested yet. However, as we noted in the response to the editor’s summary, the main goal of our study was not to test the effect of recently modified statistics. We have revised the text to make the goal of the study clearer and statements to more accurately reflect significant findings. We had chosen the word ‘anticipate’ in an attempt to make it clear that this study is not about the processing of ongoing ITD statistics but about a built-in representation of them, affecting perception. However, we have come to understand that the word ‘anticipation’ creates confusion. We have thus revised the title to “Natural ITD statistics are built into human sound localization” and revised conclusion statements to clarify this issue across the paper.

2) It is not clear how the authors computed either the ITD rate of change with azimuth (ITDrc) or ITD variability (ITDv) (they mention across-frequency interference that changes over time). Perhaps an example and equations in the methods would make this clearer.The critical contribution made by this paper is the introduction of ITD_v. The data for ITD_{rc} and ITD_v come from the LISTEN data base measured at IRCAM long ago. This database consists of physical measurements, interaural differences as a function of sound source location in free field for more than 50 individual heads. Unfortunately, it is not clear how ITD_v is determined. It is said to come from a variability in head related transfer functions (HRTFs) over time. But HRTFs do not normally depend on time. HRTFs describe wave filtering from a source in space to some chosen point in the ear. Why should they depend on time? Does ITD_v come from some estimate of the motion of listeners during the action of sound localization? Does it represent variability seen at IRCAM from one experimental session to the next for a given listener? The basis for the calculations presented here is a total mystery.The critical variable ITD_v is shown in Figure 1D. There, the variability appears to be small for small ITD values (source near the geometrical midline) and it appears to be large for large ITD values (source out to the side). However, if the temporal variation of ITD is caused by inadvertent listener rotation, then the sensitivity pattern ought to be the opposite of what appears in Figure 1D. This is so because the sensitivity of ITD to azimuth is steepest near the midline and becomes quite flat at large source azimuths.

We have made a substantial revision of the Materials and methods section explaining details of how ITD statistics were computed. We also revised Figure 1 to clarify the methods. We would like to note that ITDv was estimated as ITD variability over time but not assumed to be induced by subjects’ motion.

The basic idea is that natural acoustic signals reaching the ears are distorted in a location- and frequency-dependent manner, leading to changes in the instantaneous ITD along the duration of these signals. These changes in instantaneous ITD induce variability of ITD within short-scale time windows along sounds and across trials, affecting the natural reliability of the ITD cue. To replicate this process, we used a broadband signal, HRIRs (HRTFs impulse responses) and digital filters that reproduced cochlear filtering, as follows: (1) we convolved a broadband signal with head-related impulse responses across azimuth locations to model spatial sounds, (2) we approximated the frequency bands that reached hair cells filtering acoustic signals using parameters of human cochlear filters previously reported, and (3) we estimated ITD variability across frequency and location by computing the standard deviation of instantaneous ITD along stimulus duration.

The revised Figure 1 now provides a visual description of this method, showing that ITDv estimation for a signal from a given position in azimuth requires convolving the signal with the corresponding left- and right-ear HRIRs for each subject and passing them through model cochlear filters to estimate the standard deviation of the differences in instantaneous phase between the left and right band-filtered signals over time.

3) The manuscript is very hard to follow. The reasoning is obscure, with mathematical operations described in words instead of equations. Thus, the substance of the paper hides within a forest of details. We counted more than 120 figures. This is too many.

We are aware this manuscript addresses a complex issue, and our intention was to provide evidence of rigor of analysis with a complete reporting of data and details. However, we understand the reviewers’ concern and have streamlined the manuscript accordingly. We have added equations to the description of the methods and revised the manuscript to simplify the language. When possible, we made analysis steps more straightforward and explicative (results and conclusions did not significantly change). We have reduced the number of figures, now proposing only 4 main and 2 supplementary ones, and simplified their layout.

4) The evidence for some of the central claims of the study is either not presented sufficiently clear or tested only indirectly. We understand that the manuscript addresses an important question, and we understand the need to explain many rather complicated physical details and methodological constraints. One reviewer commended the authors for their ability to summarize and explain their data in a manner that is accessible to a broader readership. Nevertheless, the "anticipation of characteristic ITDv / statistics built into the brain" argument is not directly shown by the data that the authors provided (see next point).

We thank the reviewers for their considerate comment, which guided us in conducting a vast revision that substantially improved the manuscript. As stated above, we have revised the title and conclusions to clarify the notion of ‘anticipation’ and avoided using this wording too often. We provide specific details of revisions addressing this issue below.

5) We are not convinced that data that are presented can distinguish whether ITDv and ITDrc simply influence ITD processing or whether the brain anticipates the location-specific invariance of these statistics and exploits them to optimize ITD discrimination. I.e. the reported increase in variability at lateralized positions will certainly partially account for deterioration of JNDs (as will the decrease in ITDrc), and a large amount of data in support of this notion is presented.Does changing higher-order statistics indeed deteriorate ITD JNDs (or dITD)? The experiment summarized in Figure 4 touches on this idea (loss of ITDv due to the use of in-ear earphones results in elevated thresholds), however, as the authors noted, this threshold shift could also be (partially) due to the use of naïve listeners or other factors and thus is inconclusive. Likewise, the MMN paradigm (Figure 5) is unconvincing in this respect as well. The authors first state that the paradigm design avoids the presence of ITDv. They then show that the recorded MMN data data is best explained by a model including D-stat (i.e. ITDrc and ITDv) and argue that this finding is evidence for the anticipation of these statistics. While we agree that this finding is suggesting potentially "anticipatory processing", it is very difficult to judge the significance of the model performances without a control condition (including naturalistic free-field statistics). Some sort of comparative analysis (i.e. the manipulation of the higher-order statistics) to directly show whether neuronal processing is adapted to a specific statistical distribution would be required to justify the strong statements made in the Title, Abstract, and Introduction (last paragraph) about anticipation.

We thank the reviewers for raising these critical issues. It is true that the dITD thresholds we report obtained from our earphone testing are higher than those reported by Mills et al., (1958). An important difference, which motivated the decision to conduct our own measurements, is that Mills et al., (1958) used free-field stimulation and estimated ITD from azimuth values. Free-field stimulation, however, allows subjects to use spatial cues other than ITD to detect sound location in azimuth, which makes the lower thresholds expected. This argument is supported by the fact that ITD thresholds measured with dichotic stimulation by other previous studies (e.g., Brughera et al., 2013) are also higher than those reported by Mills. In addition, we believe our reference to using naïve listeners is important, not only to explain differences with Mills data but also to avoid the effect of training on spontaneous ITD thresholds. We have revised our justification of the differences between our data and Mills thresholds, adding this evidence. We would like to emphasize the idea that our study is not about the effect of ongoing statistics of experimental stimuli on ITD thresholds, which is why we did not attempt to manipulate them. At this time, it is difficult to predict the interplay of anticipated and actual statistics on ITD processing. Our group is currently conducting experiments examining this interaction but consider this project is beyond the scope of the present study and adding it would make it even more complex than what it already is.

[Editors' note: further revisions were suggested prior to acceptance, as described below.]

Essential revisions:1) We all regard the use of ITD statistics, specifically, the derivative (rate of change) of the mean ITD over azimuth (ITDrc) and the standard deviation of ITD (ITDv) over time as a major strength of the paper. The finding that the Stern and Colburn, (1978) and Harper and McAlpine, (2004) models can reflect a representation of ITD statistics provides a strong connection between the neural coding and psychophysics. Thus, neural density may reflect the Fisher Information for azimuthal angle changes, and the paper could be motivated by explaining why the ITD sensitivity is not equally good within the entire physiological relevant ITD range.

Thank you for this constructive suggestion. We agree with the idea that Fisher information (FI) may be a better way to represent ITDrc and ITDv statistics. We have revised the manuscript showing FI’s predictive power of our psychophysical data and its consistency with Stern and Colburn, (1978) and Harper and McAlpine, (2004) models. As recognized by reviewers, in fact the D-stat metric (ITDrc/ITDv) used in the previous submissions was closely related to Fisher information; specifically, D-stat was approximately equal to the square root of Fisher information (√FI_ITD_), and therefore conclusions were unchanged after revising the manuscript as suggested.

2) Our major concern is the wording, which often seems vague. To excerpt from a review "At the moment, the manuscript reads as if the ITD statistic is actually a detection cue, like "Oh, I now perceive the variance! The sound must come from this direction." We think you mean that fewer neurons code sound directions where there is less Fisher information (large ITDs, typically generated from large azimuthal angles) than where there is more information from the ears (small ITDs from mid-line directions). If this is what you mean, please try to be clearer and avoid phrases like terms "statistic is built in" and "the brain anticipates".

We are painfully aware that the wording did not convey our message clearly and have revised the whole manuscript to address this issue. Regarding the specific excerpt mentioned above, we must note that our paper is not conveying the idea of 'perceiving' variance and inferring from it where a sound is coming from. Rather, we are suggesting the brain possesses a prior representation of ITD statistics (ITDv and ITDrc) across frequency and location of sounds, and that this representation affects the perception and discriminability of ITD. We support the interpretation that the number of neurons coding ITD may match the natural level of Fisher information. Our paper tests that hypothesis through the Stern and Colburn and Harper and McAlpine models, showing that the distribution of neurons in these classic models of ITD discriminability are consistent with a representation of the stimulus Fisher information, such that there are fewer neurons where Fisher information is low. In this way, we argue that the human brain has a representation of ITD statistics and show that classic models of ITD coding provide a possible way that the statistics are represented.

We have extensively discussed what you mean with respect to ITD sensitivity. You imply that ITD sensitivity (and the underlying neural processing) is improved by the naturally occurring ITDv, but you don't explain how this might work, i.e. you show that the brain uses the additional variance information for ITD calculation, but whether this information necessarily follows natural statistics is not clear. Furthermore, the wording of "anticipation" implies that the brain has learned what statistics exactly define naturalistic ITDv, which is not reflected in your data. Our uncertainty about what you mean motivates point 3 below.

Thank you for the effort of helping us to improve our manuscript. We are not implying ITD sensitivity is ‘improved by the naturally occurring ITDv’ but that the brain contains a representation of ITDv that *a priori* affects ITD sensitivity. We assume that this is adaptive and creates a more efficient code for ITD because it makes ITD sensitivity finer for stimuli that are more informative about azimuth (stimuli that carry cues that vary less over stimulus duration and across trials and differ more across neighboring locations). Nor do we aim to imply that ITD ‘variance’ is ‘used’ for ITD calculation, but that ITD calculation is more accurate when ITDv is lower in natural stimuli. By using the word ‘anticipation’ in the previous version, we implied the brain has learned ITDv (and ITDrc) or contains an innate representation of it, which affects ITD perception for stimuli of different frequencies and location without needing to compute these statistics during stimulus presentation. However, our paper is not about how the statistics are learned. Our intention is to convey the idea that the brain undergoes early-life learning or, more likely, innately contains a representation of ITD statistics, including ITDv, and this is why we used the wording ‘built-in’ in earlier submissions. However, based on the critiques, we have abandoned this terminology and now refer to it as a “representation” of ITD statistics. Whether this representation of ITD statistics is genetically determined or learned is an interesting and still unanswered question, which we are currently studying. Our assumption is that the statistics that are captured over brain evolution are within a group of statistics determining Fisher information (ITDrc and ITDv).

A third point of confusion relates to D-stat. This has the highest correlation with the Mills data, and the way we understand (or interpret) your results and discussion is that even though no ITDv is present in the stimuli, the perception is explained by neural processing that incorporates the ITDv that would normally be present for these stimuli. And since the ITDv is not present, the brain must "know" (i.e. have learned) what these stats are. Please clarify if this is what you mean.

We find this paragraph quite close to what we are trying to suggest. The brain must ‘know’ and therefore has learned early in life, or has innately received, the information of the statistics underlying D-stat (ITDrc/ITDv). Here again, this is the reason why we used the ‘built-in’ terminology in the previous version of the manuscript, which was discarded in the current revision.

3) As you can tell from the above questions, our major concern is that the writing is still unclear. We have spent two weeks discussing what we think you mean. Therefore, we believe that there is potential to further clarify the wording and explanations. For example, with respect to understanding what is meant by "Natural ITD statistics are built into [the brain])", we would like to clarify whether you agree or disagree with the interpretation that "neural density may reflect the Fisher Information for azimuthal angle changes, etc" before you start revising the manuscript for a third time. Please clarify this along with your responses to the other points in a reply letter. Once we understand what you mean, we could suggest how/if to streamline the manuscript.

We feel remorse by the time our paper has taken from you and deeply appreciate the invaluable help. We agree with the interpretation that 'neural density may reflect the Fisher information for azimuthal angle changes', based on the assumption that Fisher information integrates both rate of change and variability of ITD, and this is what we are suggesting by showing that the Stern and Colburn and Harper and McAlpine papers could be consistent with our findings. By showing that the coding mechanisms underlying ITD sensitivity proposed by this paper reflects trends in natural statistics, our intention is to show how a representation of natural statistics could cause the observed effects on ITD discriminability.

Reviewer #1:This is my second review of this interesting paper. The basic idea is very compelling, that natural ITD statistics are built into the neural circuits that mediate sound localization. I also like the combination of modeling and psychophysics.Nevertheless, the presentation is quite complex. The authors have written the paper three times. Once, when first sent to the journal, prior to review, and next after the prior reviews. This version is much clearer and has less than half the number of figures.Strengths – the use of ITD statistics, specifically, the derivative (rate of change) of the mean ITD over azimuth (ITDrc) and the standard deviation of ITD (ITDv) over time are both important insights. The authors' finding, that the Stern and Colburn, (1978) and Harper and McAlpine, (2004) models can implement a representation of ITD statistics, provided a strong connection between the neural coding and psychophysics. I feel the field has been arguing about this for decades, and that this represents a major insight.

We thank the reviewer 1’s positive feedback. We have revised the manuscript extensively, further simplifying figures and reducing their number. We have also revised the text to make the main message clearer.

Reviewer #3:The authors substantially revised the manuscript by adding more detailed explanations for the calculations they made. They also employ more measured wording throughout the paper, now. I remain a bit skeptical about the suitability of their use of the terms "built in" and "anticipated" with regard to the conclusions of their findings.

We thank reviewer 3 for acknowledging the revisions in the previous submission. We have now revised the title and text across the manuscript, removing completely the terminology of ‘built in’ and ‘anticipated’ natural ITD statistics, and replacing them by saying that natural ITD statistics are ‘represented’ in the brain.

Reviewer #4:[…] Why is ITD discrimination not equally good within the entire ecologically relevant range but is better for midline directions? Because Mills' data are already converted into milliseconds of ITD, it cannot just be the geometry (e.g Kuhn, 1977). The authors argue: It is because in real life, with broad band stimulation, the interaural correlation arriving from the left and right ear at the coincidence-detecting neurons is reduced due to differing HRTF filtering. IPD and ILD fluctuations are also introduced if ITD is not perfectly compensated by internal delay line differences (e.g. differences in left and right axonal travel times). Thus, they say, why wasting large number ITD sensitive neurons to reduce neural noise on spatial locations, where spatial cues are degraded anyway (ITDv)? In addition, ITD changes per angular location change (ITDrc) are lower at these off-centre directions relative to midline directions. In other words, one could argue that the Fisher information (change of mean divided by variance – they call it D-stat) for angular changes in source direction from these direction is low and so requires less number of neurons to be coded than the far higher information from frontal directions. The so predicted density distribution of binaural detectors as a function of their best ITD can be validated with ITD discrimination experiments using single tones, as these do not get ITD (and ILD) fluctuations by going through HRTFs, auditory filters and delay lines. Because their D-stat derived with broadband stimuli fit Mill's data very well, but D-stat can itself actually not explain Mills ITD discrimination thresholds because they were obtained with pure tones and are expressed in millisecond (not azimuthal angle), one might conclude that whatever limits ITD discrimination with tones has been influenced by D-stat by experiencing everyday broadband sounds. I assume the authors imply that it is neural noise that became somehow adjusted to match D-stat. Neural noise per ITD is determined by the number of ITD coding neuron coding this ITD (e.g. Stern and Colburn, 1978; Harper and McApline, 2004).In the case that this indeed is, what the authors intend to say, I believe their idea is well worth disseminating. I agree with the first review that the current presentation is opaque and the description vague and misleading. It is lacking logical line of thoughts that tells the reader this story. The choice of wording itself is confusing and become more precise and descriptive in order to convey the message. The narrative can and should be kept simple and specific as it is actually a straightforward story, which is certainly novel and worth exploring further.

We thank reviewer 4 for the very insightful comments and the suggested connection with Fisher information. We have replaced our D-stat metric by a Fisher information index and we have revised the terminology referring to the brain representing statistics through experience and evolution. We also extensively revised the manuscript, trying to clarify the take home message and use wording easier to understand.

I would like to give further advice: The basic idea is in my opinion sufficient for an initial publication and I would not "dilute" it with unnecessary details. This story does not need further experimental data than those well-accepted ITD threshold data by Mills, (1958). They are sufficient for the purpose. Showing own, partly contradicting experimental data, just adds confusion and distracts the reader from the main message. Similarly, the EEG data are irrelevant for the narrative and just lengthen the manuscript unnecessarily.

We agree with the reviewer that the Mills, (1958) is powerful evidence supporting the goals of our study. We would like to argue that our experimental data are useful and provide more specific evidence regarding the effect of representing ITD statistics. We think showing that novelty detection is also influenced by ITD statistics is important. In addition, our measurements of dITD thresholds is relevant because it tests the hypothesis more specifically – Mills data are free field stimulation with tones, not through earphones. In fact, our data support our hypothesis rather than contradicting it. We believe that the revised figures, with removed unnecessary details, are able to show the main point of our study, supported by evidence from multiple datasets and approaches.

I agree that one can also base the neural density on the distribution of ITD tuning slope curves, assuming neurons are most sensitive to changes in ITD here (Harper and McAlpine, 2004). This is indeed worth mentioning because this thinking is becoming the popular. But the fact that an ITD tuning curve has two slopes just complicates the matter. For the sake of keeping the argument simple, I recommend not go into detail with the slope model, but illustrate the basic idea using the classic Jeffress-based model by Stern and Colburn (1978).

We agree with the notion that the two slopes of tuning curves adds complexity to Harper and McAlpine’s model prediction, which may be affecting Fisher information. We believe showing that the Harper and McAlpine, (2004) postulated coding framework could permit a representation of ITD statistics is a strong point in our manuscript, as it was pointed out by other reviewers; moreover the current analysis shows an interesting match between ITD and modeled neural population Fisher information. The models proposed in Stern and Colburn and Harper and McAlpine rely on two different frameworks explaining, respectively, psychophysics and reported tuning properties of ITD selective neurons. In our work we show the models are complementary. Furthermore, our analysis showing that adjusting the Harper and McAlpine model to match ITD statistics predicts observed neural responses suggests a potential functional connection between tuning properties and psychophysics for this model.